# HEB collaborates with TCR signaling to upregulate *Id3* and enable γδT17 cell maturation in the fetal thymus

Johanna S Selvaratnam[1,2], Juliana DB da Rocha[1,2], Vinothkumar Rajan[1,2], Helen Wang[1,2], Emily C Reddy[3], Miki S Gams[2,3], Jenny Jiahuan Liu[1,2], Cornelis Murre[4], David Wiest[5], Cynthia J Guidos[2,3], Juan Carlos Zúñiga-Pflücker[1,2], Michele Kay Anderson[1,2]*

[1]Biological Sciences, Sunnybrook Research Institute, Toronto, Canada; [2]Department of Immunology, University of Toronto, Toronto, Canada; [3]Hospital for Sick Children, Toronto, Canada; [4]Department of Molecular Biology, University of California, San Diego, San Diego, United States; [5]Blood Cell Development and Function Program, Fox Chase Cancer Center, Philadelphia, United States

*For correspondence: manderso@sri.utoronto.ca

Competing interest: The authors declare that no competing interests exist.

## eLife Assessment

The study provides **important** mechanistic insight into the transcriptional control of γδT17 development, elegantly demonstrating how HEB and Id3 act sequentially and cooperatively to regulate γδT17 cell specification and maturation. The study provides **compelling** evidence that advances the understanding of E-Id protein dynamics in thymic T cell specification. The work is comprehensive, technically rigorous, and conceptually clear, and will be of interest to immunologists, developmental biologists, and those studying the molecular underpinnings of physiological outcomes.

**Abstract** T cells expressing the γδ T cell receptor (TCR) develop in a stepwise process initiating at the αβ/γδ T cell branch point, followed by maturation and acquisition of effector functions, including the ability to produce interleukin-17 (IL-17) as γδT17 cells. Previous studies linked TCR signal strength and fate choices to the transcriptional regulator HEB (*Tcf12*) and its antagonist, Id3, but how these factors regulate different stages of γδ T cell development has not been determined. We found that immature fetal γδTCR⁺ cells from conditional *Tcf12* knockout (HEB cKO) mice were defective in activating the γδT17 program at an early stage, whereas *Id3*-deficient (Id3-KO) mice displayed a partial block in γδT17 maturation and a defect in IL-17 production. We also found that HEB cKO mice failed to upregulate *Id3* during γδT17 development, whereas HEB overexpression elevated the levels of *Id3* in collaboration with TCR signaling. Moreover, Egr2 and HEB were bound to several of the same regulatory sites on the *Id3* gene locus in the context of early T cell development. Therefore, our findings reveal an interlinked sequence of events during which HEB and TCR signaling synergize to upregulate *Id3*, which enables maturation and acquisition of the γδT17 effector program.

## Introduction

IL-17-producing γδ T (γδT17) cells are critical effectors in immune responses to bacterial and fungal pathogens and contribute to tissue repair and barrier integrity, particularly at mucosal surfaces (*Ribot et al., 2021*). Beyond host defense, γδT17 cells regulate adipose tissue homeostasis and metabolic function (*Kohlgruber et al., 2018*). However, dysregulation of γδT17 cell function has been linked to

autoimmunity and neuroinflammatory disorders, underscoring the importance of precisely controlled γδT17 cell development and function (*Alves de Lima et al., 2020*; *Papotto et al., 2018*).

While significant progress has been made in defining the stages of γδ T cell development, the transcriptional networks that govern lineage commitment and effector fate specification remain incompletely understood. In mice, γδT17 cells arise exclusively during fetal and early neonatal thymocyte development. The earliest wave of γδT17 cells in the fetal thymus, which emerges around embryonic day 17 (E17), originates from T cell precursors expressing the Vγ6 Vδ1 T cell receptor (TCR) (*Haas et al., 1993*; *Shibata et al., 2008*). A second wave of Vγ4 Vδ5 cells, which begins at E18, also gives rise to γδT17 cells (*In et al., 2017*; *Sandrock et al., 2018*). By contrast, fetal-restricted Vγ5 Vδ1 γδ T cells are associated with the IFN-γ-producing γδT1 fate (*Havran and Allison, 1990*; *Turchinovich and Hayday, 2011*). Most Vγ1 cells, which appear just before birth and continue to develop in the adult thymus, also differentiate into γδT1 cells (*Buus et al., 2017*). Although Vγ4 cells continue to be generated in the adult thymus, their capacity to give rise to innate γδT17 cells is significantly diminished (*Chen et al., 2021*; *Sandrock et al., 2018*; *Yang et al., 2023*). Instead, adult Vγ4 cells remain poised for peripheral polarization into either γδT17 or γδT1 fates (*Schmolka et al., 2013*). Other subsets, such as IL-4-producing Vγ1 Vδ6.3 cells, emerge primarily in neonatal and young mice (*Grigoriadou et al., 2003*).

γδ and αβ T cells develop primarily from shared intrathymic T cell progenitors that lack expression of CD4 and CD8 (double negative [DN]) (*Ciofani and Zuniga-Pflucker, 2010*; *Dudley et al., 1995*). These precursors can be further subdivided by CD44 and CD25 expression, with the earliest thymic immigrants classified as CD44$^+$CD25$^-$, termed early thymic progenitors (ETPs). In the fetal thymus, DN2 (CD44$^+$CD25$^+$) cells can generate either γδT17 or γδT1 cells, while DN3 (CD44$^-$CD25$^+$) cells preferentially give rise to γδT1 cells or proceed to αβ development following pre-TCR signaling in a process known as β-selection (*Dutta et al., 2021*). Successful progress through β-selection is followed by upregulation of CD8 and CD4, generating CD4$^+$CD8$^+$ double positive (DP) cells. Commitment to the γδ lineage is initiated by γδTCR signaling, which induces lineage-specifying transcription factors such as Sox13 (*Gray et al., 2013*; *Malhotra et al., 2013*), followed by effector programming factors such as Maf, which are gradually upregulated as differentiation proceeds (*Malhotra et al., 2013*; *Pokrovskii et al., 2020*). These factors promote expression of lineage-defining cytokines such as IL-17 and their regulators, including RORγt (*Zuberbuehler et al., 2019*).

TCR signal strength plays a central role in determining both the γδ T cell lineage choice and γδ T cell effector program (*Lee et al., 2010*; *Zarin et al., 2014*). Weak pre-TCR signals promote αβ lineage progression, whereas intermediate γδTCR signaling favors γδT17 cell development, and stronger signals direct γδT1 differentiation (*Haks et al., 2005*; *Zarin et al., 2015*). Signal strength is modulated by TCR affinity, proximal CD3 signaling, and cytokine crosstalk (*Fahl et al., 2018a*; *Fahl et al., 2018b*; *Michel et al., 2012*; *Muro et al., 2018*). Together, these factors converge on downstream signal transduction pathways, including the ERK-Egr-Id3 axis, which has been identified as a key mediator of TCR signal strength using manipulation of TCR ligands or TCR signaling pathways (*Bain et al., 2001*; *Fahl et al., 2018b*; *Lee et al., 2014*; *Munoz-Ruiz et al., 2016*; *Turchinovich and Hayday, 2011*).

*Id3* encodes a dominant-negative helix-loop-helix (HLH) protein that inhibits the expression of E protein-dependent genes. Id3 acts at the post-translational level by binding and sequestering E proteins, including HEB (encoded by *Tcf12*) and E2A (encoded by *Tcf3*), thereby preventing their DNA binding activity. HEB and E2A orchestrate multiple stages of αβ T cell development (*D'Cruz et al., 2012*; *Jones and Zhuang, 2011*; *Jones-Mason et al., 2012*; *Leung et al., 2025*). *Id3* expression is proportional to TCR signal strength, suggesting that it may influence T cell fate by titrating E protein activity (*Lauritsen et al., 2009*; *Zarin et al., 2018*). Moreover, *Id3* transcription is directly regulated by graded expression of Egr factors, linking TCR signal strength to E protein target gene expression (*Lauritsen et al., 2009*). We previously showed that HEB-deficient mice exhibit severe defects in γδT17 development, including impaired production of fetal Vγ4 γδ T cells and dysregulated expression of key γδT17 regulators in the Vγ6 subset (*In et al., 2017*). HEB directly activates early γδT17 genes, such as *Sox4* and *Sox13*, which are repressed when *Id3* is overexpressed. However, the complete physiological role of HEB in γδ T cell development remains unresolved.

Here, we investigated the roles of HEB and Id3 in vivo by analyzing fetal γδ T cell development in *Tcf12$^{fl/fl}$ Vav-iCre* (HEB conditional knockout [HEB cKO]) and *Id3*-deficient (Id3-KO) mice using flow cytometry and single-cell RNA sequencing (scRNA-seq) of E18 thymocytes. Our results reveal a tiered

disruption of γδ T cell development in HEB cKO mice, including dysregulated TRVG and TRDV expression, lineage diversion to the αβ program, and failure to activate key specification factors, including *Id3*. In contrast, γδT17 precursors in Id3-KO mice initiated the γδT17 specification program but failed to mature or produce IL-17. These findings suggest that HEB and Id3 function in an interlinked negative feedback loop that reinforces γδ lineage commitment and mediates the transition from specification to maturation during γδT17 cell development.

## Results

### Strategy for analyzing γδ T cell developmental progression in the fetal thymus

To analyze γδ T cell development by flow cytometry, we used a panel of antibodies that distinguish developmental stages and lineage fate choices in the mouse fetal thymus. γδ T cells develop from DN2/3 cells in the thymus from precursors with both αβ and γδ T cell potential, with upregulation of CD8 (immature single positive [ISP]) and CD4 marking commitment to the αβ-T cell lineage (*Figure 1A*). γδTe (early γδ T cell subsets prior to maturation; CD24+) cells arising from DN2/3 cells express γδTCR and CD24 on the cell surface but retain the ability to become either γδ T cells or αβ T cells (dotted arrow) (*Coffey et al., 2014*). Strong γδTCR signaling results in upregulation of CD73 (*Nt5e*) at the γδT1p stage (p=progenitor), followed by downregulation of CD24 (*Cd24a*) to yield mature CD27+CD24-CD73+CD44-γδT1 (IFNγ-producing) cells. Signaling through Vγ6 or Vγ4 TCRs results in downregulation of CD24 and upregulation of CD44 as γδ T cell precursors differentiate into immature γδT17p cells, which mature into CD27-CD24-CD73-CD44+γδT17 (IL-17-producing) cells (*Sumaria et al., 2017*). CD27 is expressed on all immature γδ T cells and stays on in mature γδT1 cells but is downregulated during γδT17 maturation. Thus, detection of γδTCR, CD4, CD8, CD24, CD73, CD27, and Vγ chains, and the genes that encode them, provides a solid framework for analyzing the impact of gene perturbations on fetal γδ T cell development and maturation.

It is important to clarify the Vγ and Vδ chain nomenclature used in this study (*Figure 1B*). Cell surface proteins detected in flow cytometry assays were classified using the Tonegawa nomenclature (*Heilig and Tonegawa, 1986*). When referring to genes or gene transcripts, the International Immunogenetics Information System (IMGT) nomenclature (*Lefranc, 2003*) was used, except in scRNA-seq figures where R-Seurat program output gene names from the Mouse Genome Informatics (MGI) site were preserved (*Baldarelli et al., 2024*). The numbering is the same in all three systems except for Vδ1, which is equivalent to TRDV4 (IMTG) and Trdv4 (MGI).

### HEB deficiency impairs Vγ4 cell development and inhibits functional maturation of γδT17 cells in the E18 fetal thymus

To investigate how HEB loss affects fetal γδ T cell development, we crossed *Tcf12*^fl/fl^ mice with *Tcf12*^fl/fl^ *Vav-iCre* mice to generate littermates without Cre (wild-type [WT]) or with Cre (HEB cKO). HEB cKO mice lack HEB in all hematopoietic cells, as previously described (*In et al., 2017*; *Welinder et al., 2011*). At E18, HEB cKO thymocytes showed reduced cellularity (*Figure 1C*) and a developmental block at the ISP-to-DP transition (*Figure 1—figure supplement 1*), as previously reported in adult HEB-deficient mice (*Barndt et al., 1999*; *Wojciechowski et al., 2007*). Although γδ T cells were proportionally increased in HEB cKO mice (*Figure 1D and E*), their absolute numbers were comparable to WT mice (*Figure 1F*), indicating that the reduced cellularity was primarily due to the loss of DP thymocytes.

We next analyzed Vγ chain subsets. Vγ6 cells were identified using an exclusion strategy as cells lacking expression of Vγ1, Vγ4, and Vγ5 (*Figure 1G and H*). It should be noted that the flow cytometry plots show the percentage of Vγ5+ and Vγ5- (Vγ6) cells out of the Vγ1-Vγ4- population (*Figure 1H*), whereas the bar graph shows the percentage of Vγ6 cells out of all γδTCR+ cells (*Figure 1I*). HEB cKO mice exhibited a marked reduction in the frequency (*Figure 1G and I*) and absolute numbers (*Figure 1J*) of Vγ4+ cells, and a corresponding increase in the other Vγ subsets (*Figure 1G, H, I, and J*). Immature (CD24+) cells were more prevalent across all Vγ subsets in HEB cKO mice, except for Vγ4+ cells, which were primarily CD24+ and thus had not yet matured, in either WT or HEB cKO mice (*Figure 1K*). Furthermore, fetal HEB cKO γδ T cells showed a major impairment in IL-17 production in response to PMA/ionomycin stimulation (*Figure 1L*), consistent with our prior fetal thymic organ

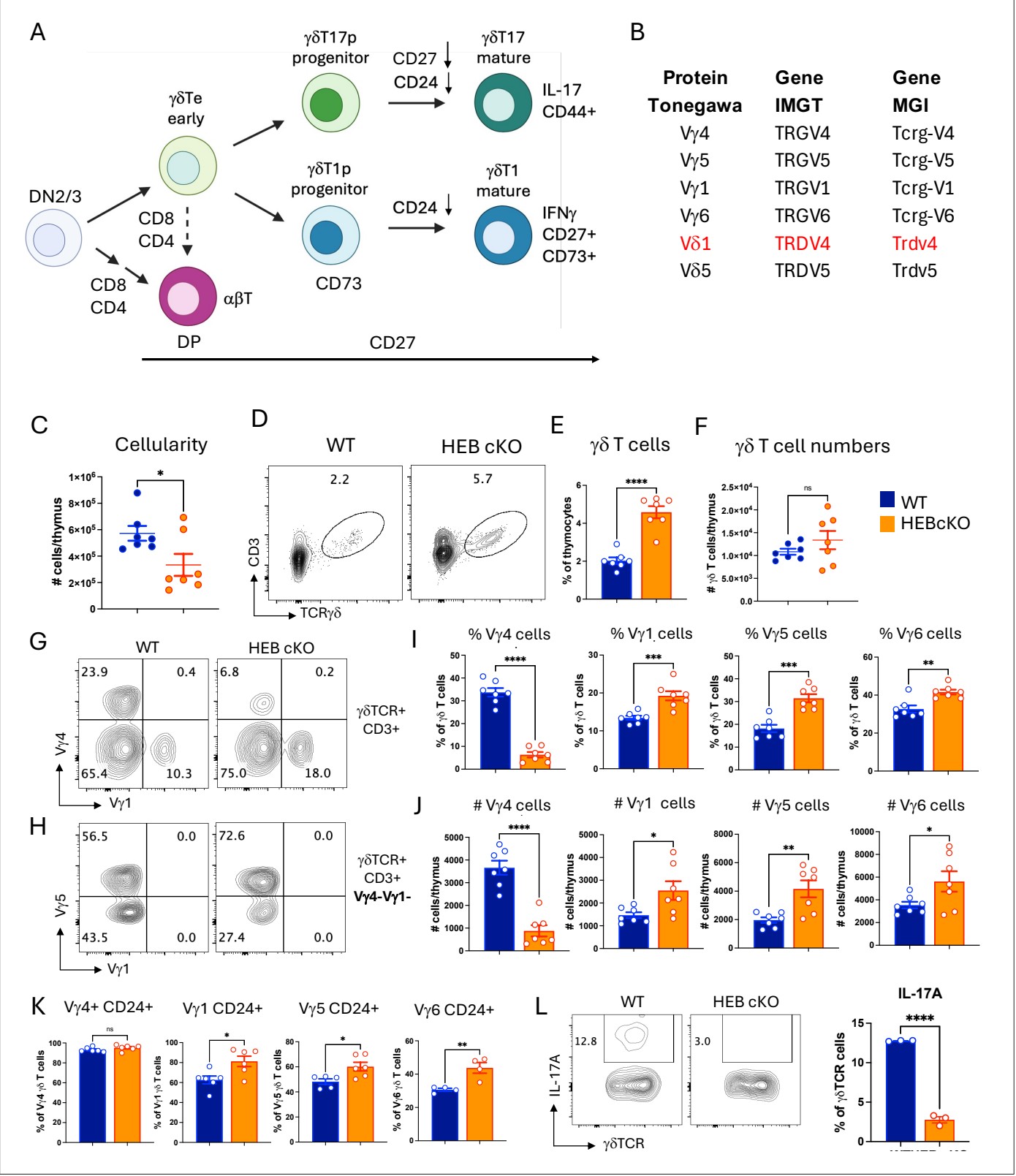

**Figure 1.** Partial block in early γδ T cell development and decrease in Vγ4 cells in HEB-deficient mice. (**A**) Stages of fetal mouse γδ T cell development. Thymocytes at the DN2 (CD4⁻CD8⁻CD44⁺CD25⁺) and DN3 (CD4⁻CD8⁻CD44⁻CD25⁺) stages of T cell development rearrange and express TCRγ, TCRδ, and TCRβ genes. DN3 cells with productive TCRβ chains expressing a pre-TCR are directed into the αβ-T cell lineage, characterized by upregulation of CD4 and CD8. Cells that express a cell surface γδTCR but have not yet committed to the γδ T cell lineage (γδTe, early γδ T cells) can still be diverted into

*Figure 1 continued on next page*

*Figure 1 continued*

the αβ T cell lineage (dotted line). γδTCR⁺ cells receiving a strong signal become CD73⁺γδT1 cell progenitors (γδT1p) that mature into IFNγ-producing γδT1 cells, whereas cells that receive an intermediate signal become γδT17 cell progenitors (γδT17p) that mature into IL-17-producing γδT17 cells. Downregulation of CD24 and CD27, and upregulation of CD44, marks maturation of γδT17 cells, whereas γδT1 cell maturation is characterized by downregulation of CD24 and maintenance of CD73 and CD27 expression. (**B**) γδ T cell nomenclature. Vγ and Vδ TCR chain genes and proteins can be identified by several different naming systems. Here, we use the Tonegawa nomenclature to refer to the proteins, and the International Immunogenetics Information System (IMGT) for the genes and transcripts. R-Seurat-generated plots use the Mouse Genome Informatics (MGI) nomenclature. The numbering for genes and proteins in these three systems are identical except for the TRDV4 gene, which encodes the Vγ1 protein (highlighted in red). (**C**) Absolute numbers of cells per thymus in wild-type (WT) (blue) and HEB conditional knockout (cKO) (orange) embryonic day 18 (E18) fetal mice. (**D, E**) Percentages of γδ T cells in WT and HEB cKO fetal thymus. (**F**) Absolute number of γδ T cells per thymus in WT and HEB cKO fetal thymus. (**G**) Flow cytometry plots of Vγ4⁺ and Vγ1⁺ cells within the γδ T cell population in WT and HEB cKO fetal thymus. (H) Flow cytometry plot of Vγ5 and Vγ6 (Vγ1⁻Vγ5⁻) expression on cells within the Vγ1⁻Vγ4⁻ population. (I) Percentages of Vγ1⁺, Vγ4⁺, Vγ5⁺, and Vγ6⁺ cells out of all γδ T cells in the WT and HEB cKO fetal thymus. (**J**) Absolute numbers of Vγ1⁺, Vγ4⁺, Vγ5⁺, and Vγ6⁺ cells per WT and HEB cKO fetal thymus. (**K**) Percentages of immature (CD24⁺) and mature (CD24⁻) γδ T cells out of all γδ T cells in each Vγ subset in WT and HEB cKO fetal thymus. (**L**) Expression of IL-17A protein in γδ T cells from E18 thymus stimulated with PMA/ionomycin as assessed by intracellular staining. Experiments were done two to three times, and results were pooled for analysis. Each biological replicate is depicted as an open circle on the bar graphs. Blue = WT, orange = HEB cKO. Significant differences between WT and HEB cKO subsets were determined using unpaired classic Student's t-tests. *p<0.05, **p<0.01, ***p<0.001, ****p<0.0001.

The online version of this article includes the following figure supplement(s) for figure 1:

**Figure supplement 1.** Defects in αβ T cell development in embryonic day 18 (E18) fetal thymus of HEB conditional knockout (cKO) mice.

---

culture results (*In et al., 2017*). These findings indicate that even in the intact E18 thymic environment, HEB is essential for γδT17 cell development.

## Single-cell transcriptomic analysis reveals a role for HEB in establishing γδ T cell identity

To assess the transcriptomic impact of HEB deficiency, we performed scRNA-seq on γδTCR⁺ thymocytes sorted from E18 WT and HEB cKO littermates. After quality control, we merged the datasets, excluded myeloid cells, and regressed out cell cycle genes using Seurat (*Hao et al., 2021*). Our analysis identified eight distinct clusters (numbered 0–7) visualized as a UMAP (*Figure 2A*). WT cells were enriched in clusters 2 and 4, while HEB cKO cells dominated clusters 0, 1, and 3, whereas clusters 5–7 showed a more balanced distribution (*Figure 2B and C*). To annotate the clusters, we curated 90 γδ T cell subset-defining genes from prior studies (*Inácio et al., 2025*; *Liu et al., 2020*; *Mistri et al., 2024*; *Narayan et al., 2012*; *Pokrovskii et al., 2020*; *Spidale et al., 2018*; *Yang et al., 2023*; *Figure 2—source data 1*). The top 10 differentially expressed genes were used to generate a clustered dot plot (*Figure 2D*), which we used to identify each cluster. We classified clusters 2 and 6 as early γδT cells (randomly assigned as γδTe1 and γδTe2), clusters 1 and 4 as γδT17 progenitors (γδT17p), cluster 0 as γδT1 progenitors (γδT1p), cluster 7 as mature γδT1 cells (γδT1), cluster 5 as mature γδT17 cells (γδT17), and cluster 3 as αβ lineage-like cells (αβT).

Quantification of WT and HEB cKO cells per cluster (*Figure 2E*) revealed that the γδT17p clusters segregated by genotype into WT (γδT17pw) and HEB cKO (γδT17pk) cells. HEB cKO cells were also over-represented in the αβT cluster and under-represented in the γδTe1 cluster. An unbiased heatmap of the top 10 most differentially expressed genes across all clusters further validated our assigned identities and revealed additional genes associated with these subsets (*Figure 2F*). These findings suggest that the loss of HEB impairs the early γδ T cell program and allows diversion toward the αβ T cell lineage.

## HEB deficiency suppresses TRDV5 expression and promotes TRDV4 expression

We next confirmed that WT and HEB cKO cells within each cluster represented equivalent developmental subsets (*Figure 2—source data 1*). All immature γδ T cell clusters expressed *Cd24a* and *Cd27*, while mature subsets lacked *Cd24a* (*Figure 2—source data 1*). Among mature cells, only γδT1 cells expressed *Cd27* and *Nt5e* (CD73), whereas γδT17 cells were uniquely *Cd44*-positive, consistent with our assignments. We also analyzed TRGV and TRDV gene expression (*Figure 2—figure supplement 1B and C*). In WT cells, TRGV4 and TRDV5 were co-expressed in γδTe and γδT17p populations but were greatly diminished in HEB cKO cells. TRGV6 and TRGV5 transcripts were detectable in γδTe

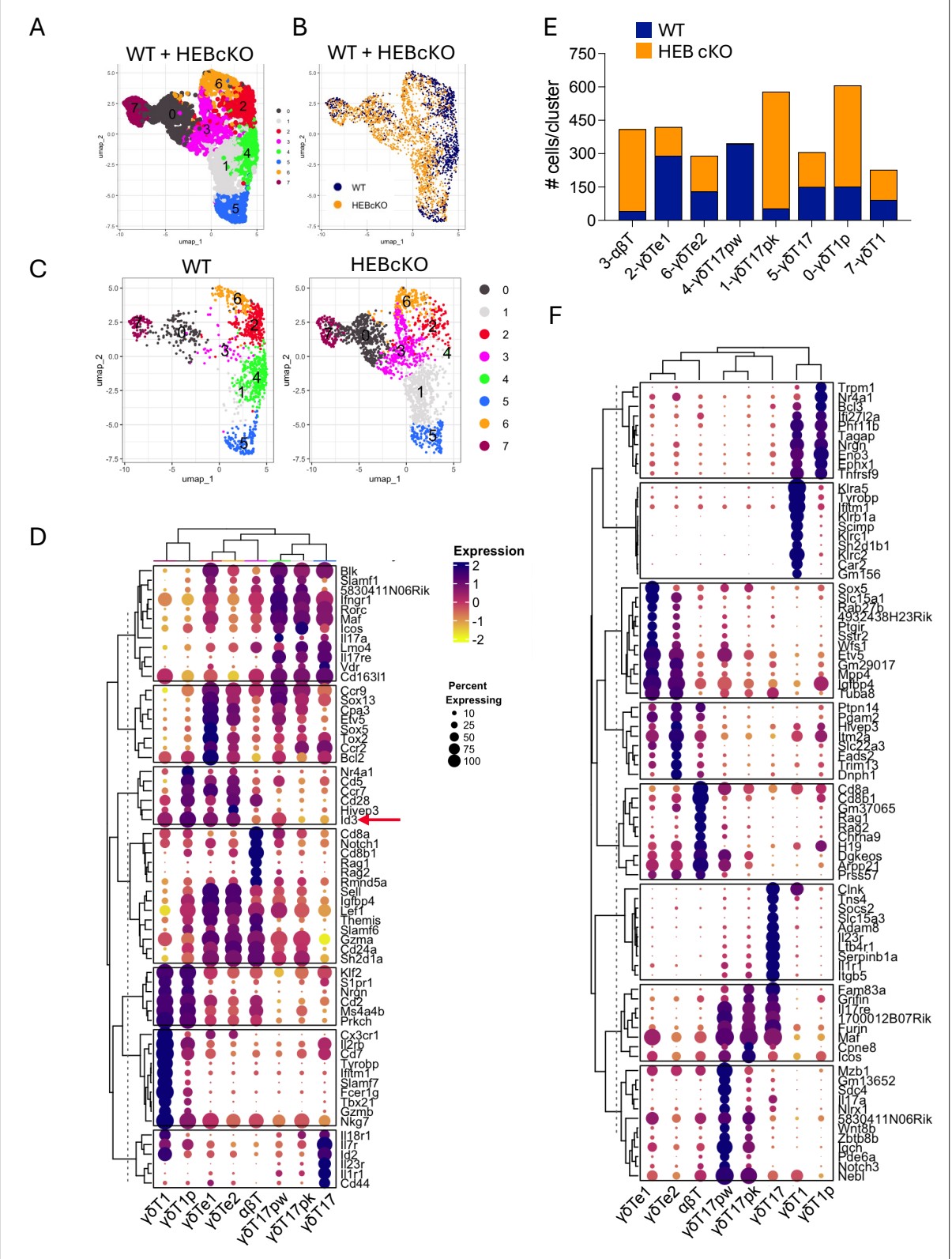

**Figure 2.** Identification of γδ T cell subsets in wild-type (WT) and HEB conditional knockout (cKO) fetal thymus by single-cell RNA sequencing (scRNA-seq). γδ T cells were sorted from embryonic day 18 (E18) fetal thymuses from WT (3) or HEB cKO (3) mice, pooled according to genotype, and subjected to scRNA-seq and analysis. (**A**) Uniform manifold approximation and projection (UMAP) plots depicting merged WT and HEB cKO cells in eight clusters (0–7). (**B**) Grouped UMAP showing the distribution of WT (blue) and HEB cKO (orange) cells across all clusters. (**C**) Split UMAP plots showing

*Figure 2 continued on next page*

*Figure 2 continued*

the distribution of cells in WT (left) and HEB cKO (right) clusters; note that cluster 4 is restricted to WT cells, and cluster 1 is heavily biased toward HEB cKO cells. (**D**) Genes previously identified as signatures for developmental and functional γδ T cell subsets were compiled from previously published reports. The top 10 most differentially expressed genes from this list were visualized as a clustered dot plot, which was used to assign cluster identities. Two clusters corresponding to early γδ T cells were randomly designated as γδTe1 and γδTe2. (**E**) Numbers of WT (blue) and HEB cKO (orange) cells per cluster. (**F**) Unbiased clustered dot plot of the top 10 most differentially expressed genes across all clusters. In the clustered dot plots, the percentage of cells expressing the gene in each cluster is depicted by the size of the dot, and the color indicates the relative magnitude of expression across clusters.

The online version of this article includes the following source data and figure supplement(s) for figure 2:

**Source data 1.** References for curated gene list used to identify differentially expressed genes and assign cluster identities in *Figures 2 and 6*.

**Figure supplement 1.** TRGV and TRDV expression profiling reveals depletion of TRGV4 and TRDV5 transcripts and overexpression of TRDV4 in the HEB conditional knockout (cKO) fetal γδ T cells.

---

subsets from HEB cKO mice, consistent with a delay in Vγ6 and Vγ5 γδ T cell maturation, whereas TRDV4 was expressed more broadly and at higher levels in all immature HEB cKO γδ T cell subsets relative to WT counterparts. These results indicate that HEB plays an important role in maintaining the subset specificity and magnitude of TRGV5 and TRDV4 expression during γδ T cell development.

## HEB deficiency impairs early γδ T cell signatures and enhances αβ-T lineage features

To further define differences between WT and HEB cKO cells at the transcriptomic level, we constructed gene modules composed of suites of signature genes for the γδTe1, γδTe2, αβT, γδT17p, γδT17, γδT1p, and γδT1 subsets, derived from the analysis of merged WT and HEB cKO cells (*Figure 2D and F*; see Materials and methods). These modules were used to assign scores in WT (left) and HEB cKO (right) cells, which were visualized using split dot plots.

This analysis revealed a pronounced loss of the γδTe1 gene signature in HEB cKO cells (*Figure 3A*), alongside a notable increase in the αβT signature (*Figure 3B*). The γδTe2, γδT17p, and γδT17 gene signatures were also diminished in HEB cKO cells relative to WT, whereas the γδT1p signature was enriched. We also examined the expression of individual genes that were diagnostic for γδ T cell subsets (*Figure 3C*) versus committed αβ T cells (*Figure 3D*). In the γδTe1 and γδTe2 subsets of HEB cKO mice, *Sox13* and *Etv5* were reduced and *Cd8b1* and *Dgkeos* were elevated relative to their WT counterparts. In γδT17p cells, αβT lineage genes were not detectable, but *Sox13* and *Etv5* were still lower in HEB cKO cells than WT cells, potentially decoupling the specification and commitment events. A less dramatic reduction in *Il1r1* was observed in HEB cKO γδT17 cells (*Figure 3E*), and the levels of *Nrgn* and *Eomes* were similar in γδT1 lineage cells (*Figure 3F*). These results indicated that the impact of HEB deficiency at the transcriptomic level was more pronounced in early γδ T cell subsets than in mature γδ T cells.

## HEB deficiency obstructs γδT17 progenitor development and dampens TCR signal strength

The γδT17p subset segregated into distinct clusters in WT and HEB cKO cells, highlighting a critical stage of HEB-dependent regulation. To explore this further, we performed an unbiased differential gene expression analysis in WT cells versus HEB cKO cells and visualized the results using an enhanced volcano plot with stringent thresholds ($\log_2$FC>0.5, $-\log_{10}$p>25) (*Figure 4A*). Many of the top differentially expressed genes were signature markers of γδT17 differentiation, including *Blk*, *Sox13*, and *Etv5*, all of which were downregulated in HEB cKO cells. *Trdv4* was among the most upregulated genes in HEB cKO γδT17p cells.

We next generated a list of differentially expressed genes between WT and HEB cKO γδT17p cells using less stringent criteria (adj p-value>0.001, $\log_2$FC >0.25) (*Figure 4—source data 1*). This list was subjected to gene ontology analysis using ShinyGO, which resulted in a list of Kyoto Encyclopedia of Genes and Genomics (KEGG) pathways with reduced gene representation in HEB cKO γδT17p cells. We observed significant depletion of gene pathways related to TCR signaling (PD-1 checkpoint, NF-κB signaling) and Th1, Th2, and Th17 differentiation in HEB cKO cells relative to WT cells (*Figure 4—source data 2*). Three key markers of TCR signal strength, *Cd5*, *Cd69*, and *Egr1*, were reduced in both γδTe2 and γδT17p subsets in HEB cKO mice (*Figure 4C*). *Id3* expression was markedly decreased

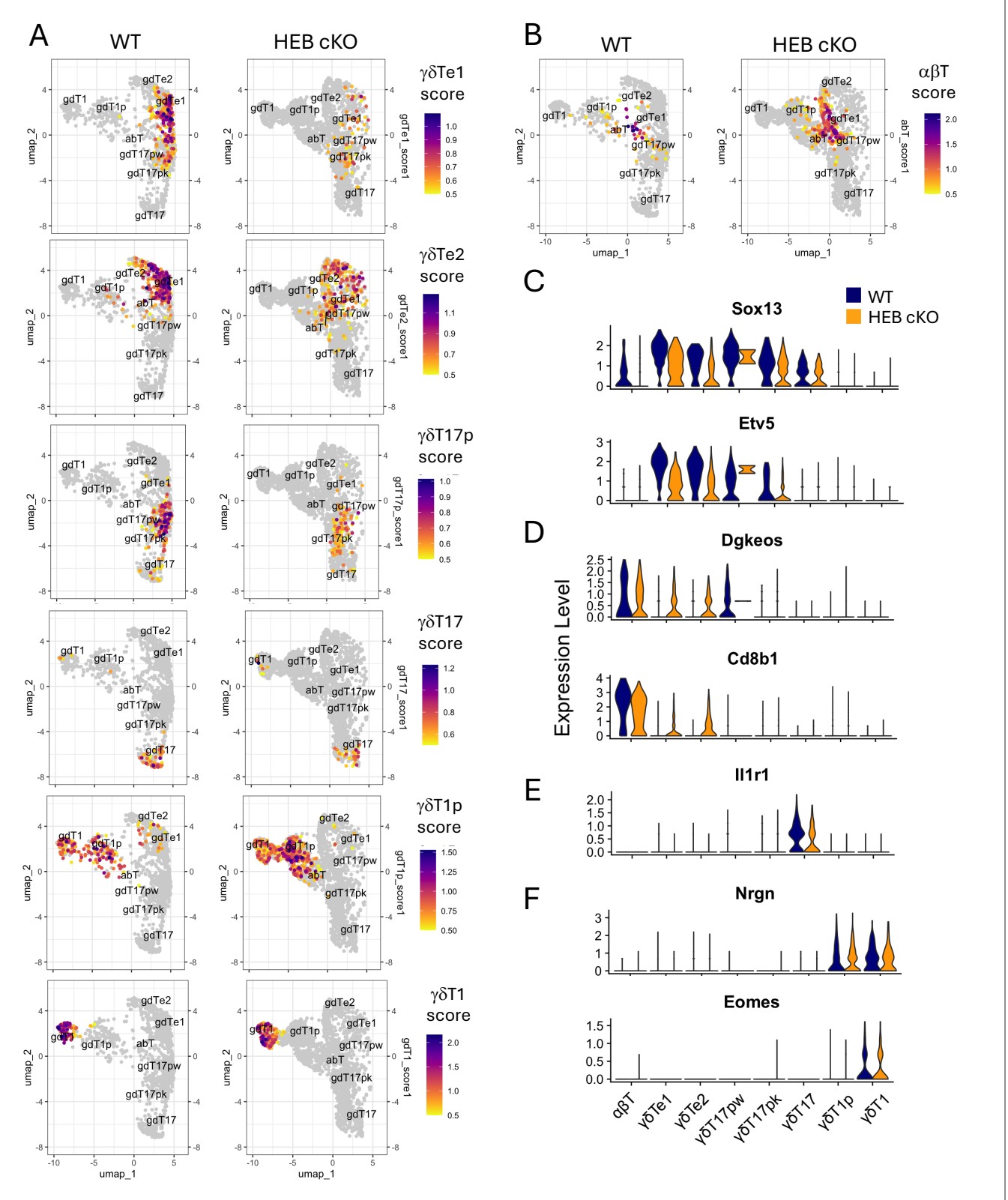

**Figure 3.** The αβ T gene program is expanded, and the γδT17 precursor gene program is lost in HEB conditional knockout (cKO) cells. γδ T cells sorted from embryonic day 18 (E18) fetal thymuses from wild-type (WT) and HEB cKO mice were subjected to single-cell RNA sequencing (scRNA-seq) and analysis. (**A, B**) Gene modules were generated from subset-biased genes, and cells were scored for each module. Module scores are depicted as split feature plots, with wild-type (WT) plots on the left and HEB cKO plots on the right. Module scores that characterize γδ T cell subsets are shown in

*Figure 3 continued on next page*

Figure 3 continued

(**A**), and a module score for the αβ-T lineage is shown in (**B**). (**C–F**) Split violin plots of genes that typify different γδ T cell subsets as follows: (**C**) γδTe/γδT17p cells, (**D**) αβ T cells, (**E**) γδT17 cells, (**F**) γδT1p and γδT1 cells. Blue = WT, orange = HEB cKO.

in HEB cKO γδTe cells and nearly absent in γδT17p cells but remained intact in γδT1p and γδT1 cells (*Figure 4D and E*). We also examined the expression of *Maf* and *Rorc*, two key regulators of γδT17 maturation, and found that they were expressed similarly between HEB cKO and WT cells in each cluster (*Figure 4F*). Therefore, HEB modulates TCR signaling during γδT17 specification but appears to be largely dispensable for the expression of γδT17 cell maturation factors.

## *Id3* expression in γδT17 progenitors is controlled through HEB-dependent mechanisms

To further elucidate the landscape of E protein and Id protein expression during γδ T cell development, we examined the expression of *Tcf12* (encodes HEB), *Tcf3* (encodes E2A), *Id3*, and *Id2* in each cluster (*Figure 4—figure supplement 1*). *Id1* and *Id4* transcripts were not detectable in any of our datasets. It should be noted that the *Tcf12* deletion occurs in one of the last exons of a 200 kb gene locus, and although the protein is absent (*Barndt et al., 1999*), some mRNA expression can still be detected. In WT mice, γδTe cells co-expressed *Tcf12*, *Tcf3*, and *Id3* (*Figure 4—figure supplement 1A–C, E*), consistent with dynamic regulation of E protein-dependent target gene expression during the αβ/γδ fate choice (*Murre, 2019*). γδT17p cells co-expressed *Tcf12* and *Tcf3*, whereas γδT1 cells co-expressed *Tcf3* and *Id3* (*Figure 4—figure supplement 1B and D*). *Id2* expression was restricted to mature γδT17 and mature γδT1 cells in both WT and HEB cKO mice. *Tcf3* expression was also unaffected by a lack of HEB, indicating that *Tcf3* and *Id2* expression are HEB-independent in this context (*Figure 4—figure supplement 1E and F*). *Id3* expression was disrupted in HEB cKO cells in γδTe and γδT17 cells but not in γδT1p cells (*Figure 4—figure supplement 1C and D*). This analysis reveals that *Id3* expression was disrupted only in cells that normally express *Tcf12*.

## Loss of *Id3* impairs CD73 upregulation and γδT17 cell function in the fetal thymus

Given the clear dependence of *Id3* expression on HEB in specific γδ T cell subsets, we next investigated how loss of *Id3* itself affects γδ T cell development using E18 fetal thymocytes from *Id3* knockout (Id3-KO) mice. Total thymic cellularity in Id3-KO mice was comparable to WT controls (*Figure 5A*), but the proportion of γδ T cells among total thymocytes was reduced (*Figure 5B and C*). Analysis of Vγ chain usage revealed no major differences between WT and Id3-KO mice, although there was a slight increase in the proportion of Vγ6 cells in Id3-KO γδ T cells (*Figure 5E and F*). Interestingly, very few Id3-KO γδ T cells expressed CD73 (encoded by *Nt5e*) (*Figure 5G and H*), including Vγ5 and Vγ1 cells (*Figure 5—figure supplement 1*). *Nt5e* expression is upregulated in response to strong TCR signaling during γδ T cell development, and it can also be induced in response to short-term stimulation (*Buus et al., 2016*; *Coffey et al., 2014*). To determine whether the deficiency in CD73 expression in Id3-KO mice reflected an expansion of CD73⁻ cells or a failure to induce *Nt5e*, we cultured E18 fetal thymocytes from WT and Id3-KO mice with PMA/ionomycin (P/I) for 4 hr and measured CD73 expression in γδ T cells by flow cytometry (*Figure 5I and J*). While CD73 was robustly induced in a substantial fraction of CD27⁺ γδ T cells from WT mice, it remained nearly undetectable in Id3-KO γδ T cells, indicating a direct role for Id3 in the regulation of *Nt5e* expression.

Mature CD73⁻ γδ T cells typically exhibit a bias toward the γδT17 lineage (*In et al., 2017*). We hypothesized that loss of *Id3* might re-direct γδT1-fated cells into the γδT17 lineage. However, upon stimulation, Id3-KO γδ T cells failed to produce IL-17 (*Figure 5K and L*), indicating that Id3 deficiency does not promote γδT17 differentiation. These findings highlight a critical role for *Id3* in the functional maturation of both CD73⁺ and CD73⁻ γδ T cells.

## Loss of Id3 disrupts expression of γδT17 cell maturation transcription factors

To further investigate how Id3 deficiency affects γδ T cell development, we measured intracellular expression of PLZF (encoded by *Zbtb16*) and MAF in E18 γδ T cells from WT and Id3-KO mice by

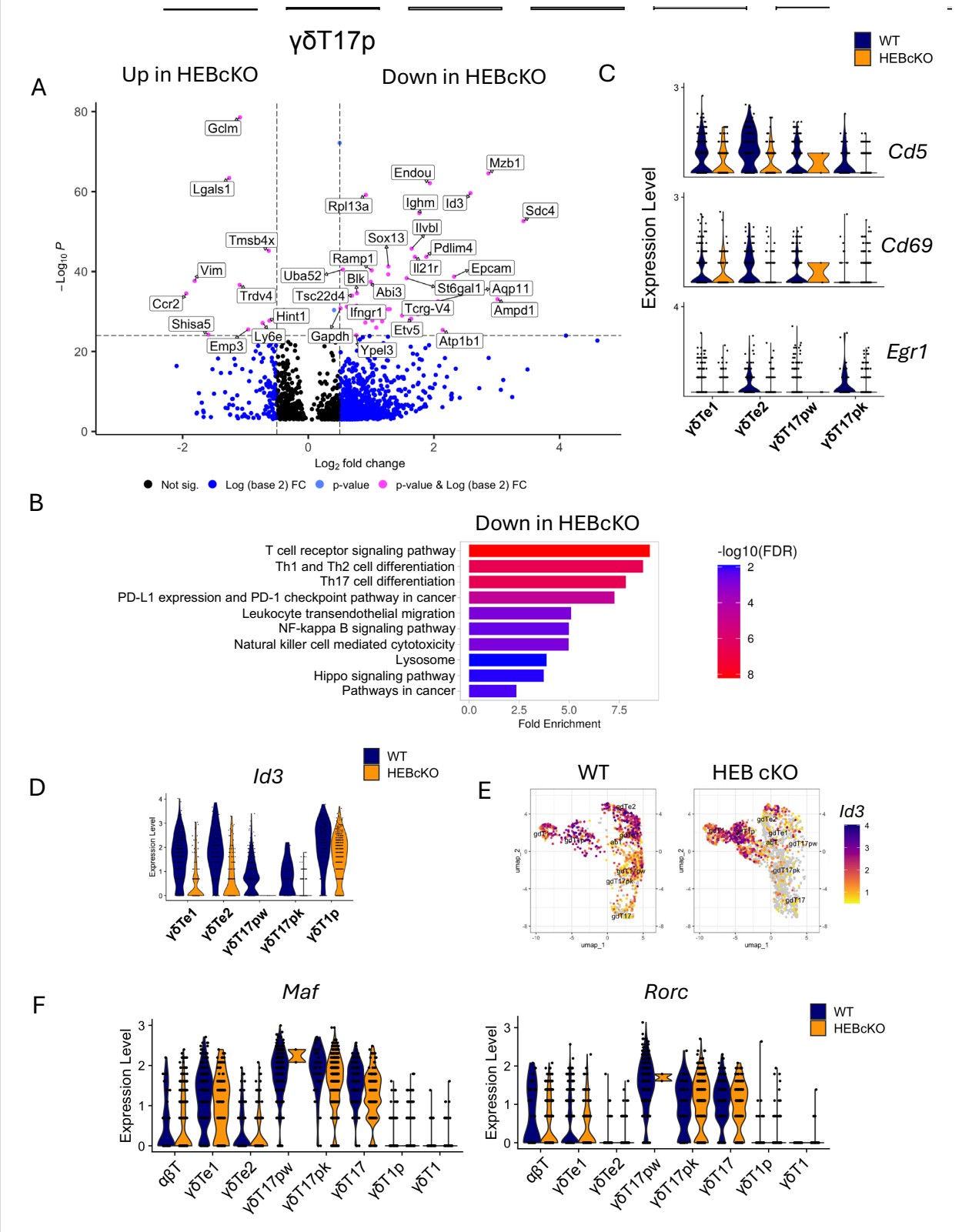

**Figure 4.** Decreases in T cell effector differentiation and T cell receptor (TCR) signaling genes in γδT17p cells from HEB conditional knockout (cKO) mice. (**A**) Volcano plots showing differential gene expression in γδT17p cells from wild-type (WT) versus HEB cKO fetal thymus. Genes expressed at higher levels in HEB cKO cells are on the left, and genes expressed at lower levels in HEB cKO cells are on the right. Significance (pink) was set at log$_2$FC>0.5 and –log$_{10}$p<10$^{25}$. (**B**) Gene ontology analysis of genes significantly reduced in HEB cKO γδT17p cells relative to WT, with significance set at

*Figure 4 continued on next page*

*Figure 4 continued*

avg log$_2$FC>0.25 and adjusted p-value<0.001. Bar plots show pathway enrichment (fold enrichment) and significance by false discovery rate (FDR) for each functional category defined in the Kyoto Encyclopedia of Genes and Genomics (KEGG) pathway list. Minimum genes for pathway inclusion was set at 5, and FDR cutoff was set at 0.05. (**C**) Relative expression of genes associated with strong TCR signaling in WT and HEB cKO cells in each cluster. (**D**) Relative expression of *Id3* in immature γδ T cell subsets from WT and HEB cKO mice. (**E**) Split feature plots showing expression of *Id3* across all clusters in WT versus HEB cKO cells. (**F**) Relative expression of *Maf* and *Rorc* in WT versus HEB cKO γδT cell subsets. WT = blue, HEB cKO = orange.

The online version of this article includes the following source data and figure supplement(s) for figure 4:

**Source data 1.** Differentially expressed genes between wild-type (WT) and HEB conditional knockout (cKO) γδT17 precursor populations from *Figure 4A*.

**Source data 2.** Enriched Kyoto Encyclopedia of Genes and Genomics (KEGG) pathways and gene members between wild-type (WT) and HEB conditional knockout (cKO) γδT17 precursor populations, as shown in *Figure 4B*.

**Figure supplement 1.** Patterns of E protein and Id protein gene expression during γδ T cell development in wild-type (WT) and HEB conditional knockout (cKO) mice.

flow cytometry (*Figure 6*). PLZF is expressed in innate fetal/neonatal γδ T cells and adult iNKT and IL-4-producing γδ T cells (*Alonzo et al., 2010*; *Kreslavsky et al., 2009*; *Lu et al., 2015*). To compare developmental stages within Vγ subsets between WT and Id3-KO γδ T cells, we gated on: (1) immature (CD24$^+$) Vγ4 cells, which comprised the majority of Vγ4 cells, (2) immature (CD24$^+$) Vγ6 cells, and (3) mature (CD24$^-$) Vγ6 cells (*Figure 6A*). This analysis yielded three populations, based on PLZF and MAF expression: PLZF$^+$MAF$^+$, PLZF$^+$MAF$^-$, and PLZF$^-$MAF$^-$ cells (*Figure 6A and B*).

In WT mice, most immature Vγ4 and Vγ6 cells expressed PLZF, with about half of the PLZF$^+$ cells co-expressing MAF. Nearly all mature Vγ6 cells co-expressed PLZF and MAF. Immature Vγ4 and Vγ6 subsets in Id3-KO mice had reduced proportions of PLZF$^+$MAF$^+$ cells and increased percentages of PLZF$^-$MAF$^-$ cells relative to WT. Mature Vγ6 cells in Id3-KO mice exhibited a more severe disruption, with only ~50% co-expressing PLZF and MAF, and significant increases within the PLZF$^+$MAF$^-$ or PLZF$^-$MAF$^-$ subsets. Moreover, mean fluorescence intensity analysis revealed substantially lower PLZF protein levels in Id3-KO cells across all subsets, especially immature cells, suggesting that this defect is not solely due to delayed maturation (*Figure 6C*).

## Id3 deficiency promotes the αβ-T lineage but does not disrupt the expression of early γδ T cell regulators

To investigate population dynamics and gene expression changes in Id3-KO fetal thymocytes, we performed scRNA-seq on CD4$^-$CD8$^-$ E18 fetal thymocytes. This strategy was designed to capture γδ T-biased cells with low surface γδTCR that might be missed by flow cytometric cell sorting. After quality control, WT and Id3-KO datasets were merged and subjected to dimensionality reduction, which identified 11 clusters (0–10) (*Figure 6—figure supplement 1A*). Violin plots of lineage and subset-specific genes identified two γδ T cell clusters (*Figure 6—figure supplement 1B*), which were computationally isolated and re-clustered. This process yielded four new γδ T cell clusters (0–3), which we identified as γδTe, γδT17p, γδT17, and γδT1 cells (*Figure 6D and E*), using the same strategy shown in *Figure 2* (*Figure 6G*). This scRNA-seq dataset lacked a γδ/αβ T lineage cluster, likely due to low cell numbers and/or exclusion of CD4$^+$ and CD8$^+$ cells in the enrichment strategy. However, flow cytometry of E18 fetal thymocytes revealed a significantly higher proportion of TCRγδ$^+$ cells co-expressing CD4 and CD8 in Id3-KO mice than in WT mice, indicating a bias toward the αβ T cell program (*Figure 6—figure supplement 2*). Notably, our scRNA-seq analysis showed that γδ T cell specification genes (*Sox13*, *Etv5*) and *Tcf12* were unaffected by the decrease in *Id3* (*Figure 6H*). These results decouple the loss of *Id3* from the disruption of early γδ T cell regulators in HEB cKO mice.

## *Id3* is required for the induction of γδT17 regulators at the transcriptional level

Compared to WT, the Id3-KO mice had increased numbers of γδT17p cells and fewer mature γδT17 cells (*Figure 6F*), consistent with a partial block in γδT17 maturation. *Id2* expression was markedly increased in all Id3-KO γδ T cell subsets (*Figure 6I*), consistent with a previously reported compensatory role (*Zhang et al., 2014*). *Maf* and *Zbtb16* transcripts were markedly reduced in Id3-KO cells at the γδTe stage but recovered to near WT levels in mature γδT17 cells (*Figure 6I*). We noted a similar

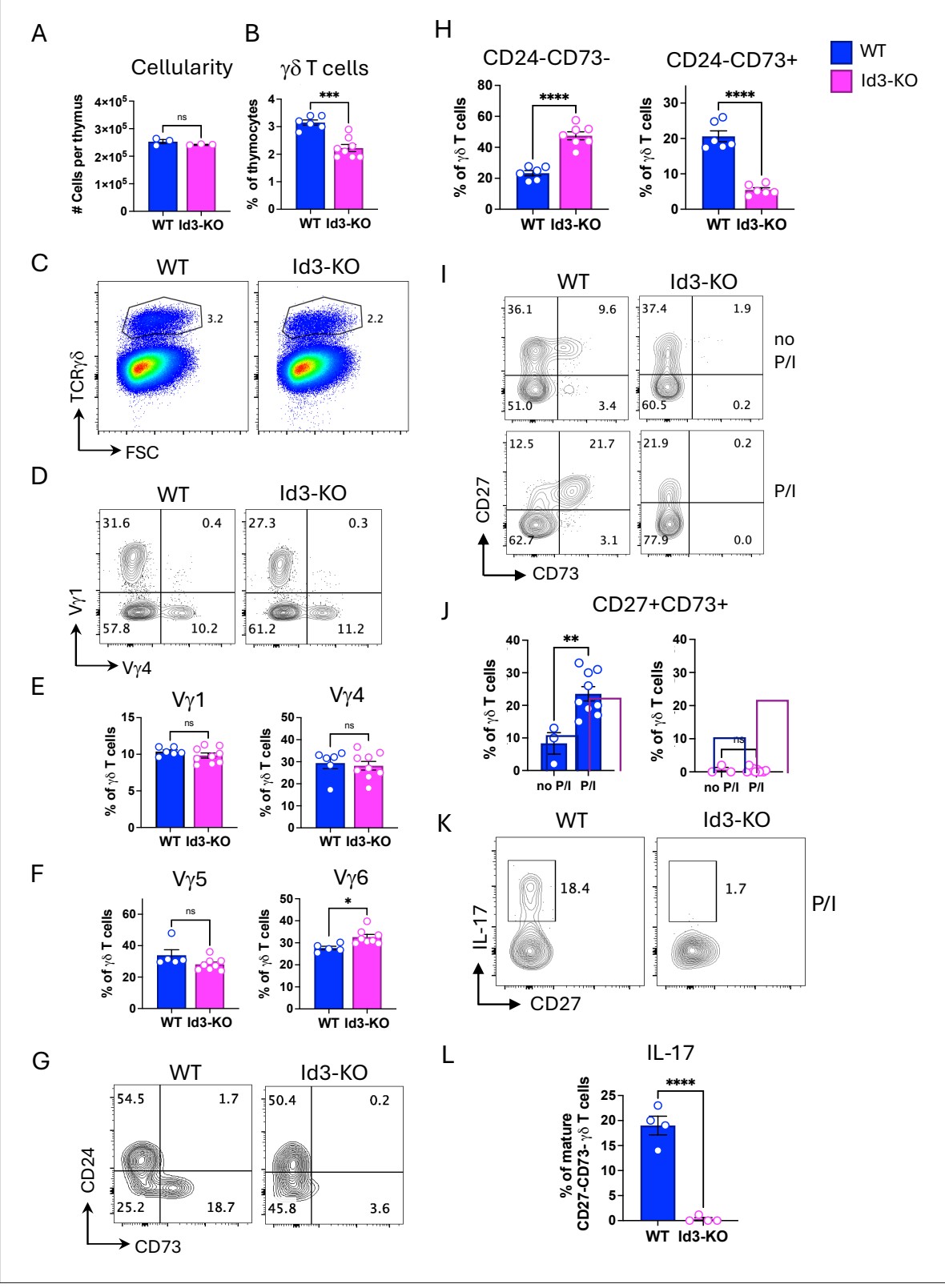

**Figure 5.** Fetal γδ T cells from Id3-KO mice are defective in CD73 upregulation and interleukin-17 (IL-17) production. (**A**) Absolute numbers of cells per embryonic day 18 (E18) fetal thymus from wild-type (WT) and Id3-KO littermate mice. (**B, C**) Quantification (**B**) and flow cytometry plots (**C**) of the percentages of γδ T cells out of all thymocytes. (**D, E**) Flow cytometry plots (**D**) and quantification (**E**) of Vγ1+ and Vγ4+ out of all γδ T cells. (**F**) Percentages of Vγ5+ and Vγ6+ out of all γδ T cells. (**G**) Flow cytometry plots of CD24 and CD73 expression in γδTCR+ cells. (**H**) Quantification of

*Figure 5 continued on next page*

*Figure 5 continued*

mature (CD24⁻) CD73⁺ and CD73⁻ γδ T cells out of all γδ T cells. (**I**) Flow cytometry plots of expression of CD27 and CD73 expression in unstimulated (top) and stimulated (bottom) γδ T cells. (**J**) Percentages of CD27⁺CD73⁺ cells out of all γδ T cells under unstimulated or stimulated conditions. (K, L) Flow cytometry (**K**) and quantification (**L**) of the percentages of CD27⁻CD73⁻CD24⁻ (primarily mature Vγ6) cells expressing IL-17 in response to stimulation. Experiments were done two to three times, and results were pooled for analysis. Each biological replicate is depicted as an open circle on the bar graphs. Blue = WT, pink = Id3 KO. P/I=phorbol 12-myristate 13-acetate (PMA)+ionomycin. Significant differences between WT and Id3-KO subsets were determined using unpaired classic Student's t-tests. *p<0.05, **p<0.01, ***p<0.001, ****p<0.0001.

The online version of this article includes the following figure supplement(s) for figure 5:

**Figure supplement 1.** CD73 is upregulated during development of Vγ5 and Vγ1 γδ T cells in wild-type (WT) but not Id3-KO fetal thymus.

impact on expression of *Rora* in Id3-KO mice. *Rora* is not required for γδT17 cell function (**Barros-Martins et al., 2016**), but does play roles in the development of Th17 (**Hall et al., 2022**) and ILC3 cells (**Lo et al., 2016**). We also examined expression of *Rorc* (encodes RORγt) (**Figure 6—figure supplement 1**). Given the decrease in other genes involved in γδT17 cell maturation and the dependence of *Rorc* expression on MAF in γδT17 cells (**Zuberbuehler et al., 2019**), we were surprised to find that *Rorc* was higher in Id3-KO γδT17p cells than in WT γδT17p cells (**Figure 6—figure supplement 1D**) and was also elevated in the *Cd8a*-expressing DN4 subset (**Figure 6—figure supplement 1E**). Although RORγt is a critical regulator of γδT17 development and function, it is also upregulated after β-selection and maintains DP cell survival (**Xi et al., 2006**), suggesting that the increase we observed in the γδT17p cells could in part be due to αβ-T cell lineage diversion.

## ChIP-seq analysis reveals shared HEB, E2A, and Egr2 binding sites in the *Id3* locus

Although our data showed that HEB is necessary for *Id3* expression during γδT17 development, it remained unclear whether HEB directly regulates *Id3*. The *Id3* gene locus is composed of three exons adjacent to the long non-coding RNA gene *Gm42329*, which lies in the opposite orientation (**Figure 7A**). To assess direct binding of HEB and/or E2A to the *Id3* locus, we analyzed ChIP-seq datasets from *Rag2⁻/⁻* DN3 thymocytes (**Fahl et al., 2021**). This analysis identified three major binding regions for both HEB and E2A upstream of the first exon of *Id3*, each containing multiple peaks. We also examined ChIP-seq datasets from DN3 cells stimulated with anti-CD3 or anti-TCRβ to mimic TCR signaling (**Pekowska et al., 2011**; **Seiler et al., 2012**). RNA polymerase II bound the *Id3* promoter in *Rag2⁻/⁻* mice stimulated with anti-CD3ε, revealing active *Id3* transcription in cells that had experienced CD3-mediated signaling. Additionally, we observed binding of Egr2 to two sites that overlapped with the HEB/E2A bound regions in total thymocytes from mice that had been injected with anti-TCRβ (**Seiler et al., 2012**). Notably, H3K27me3 repressive marks were detectable across the *Gm42329* locus but were absent from *Id3*, indicating that *Id3* is epigenetically poised for activation before pre-TCR and/or γδTCR signaling occurs.

Given that the data from the Egr2 and HEB/E2A ChIP-seq analyses were generated using different thymocyte subsets, we analyzed additional ChIP-seq data for HEB and E2A binding in *Rag2⁻/⁻* DN3 cells that had been transduced with retroviral constructs encoding the KN6γδTCR and cultured with stroma expressing the weak KN6 ligand T10 for 4 days (**Fahl et al., 2021**). These *Rag2⁻/⁻* DN3 + γδTCR cells allowed us to examine HEB/E2A binding to sites in the *Id3* locus in a cellular context more closely aligned to the Egr2 binding assay (**Figure 7—figure supplement 1A**). This analysis revealed that the binding of HEB/E2A on those sites persisted after weak γδTCR signaling, strengthening the likelihood that concurrent binding of HEB/E2A and Egr2 occurs during this developmental transition. We noted that HEB/E2A binding was slightly dampened in *Rag2⁻/⁻* DN3 + γδTCR cells relative to *Rag2⁻/⁻* DN3 cells, consistent with the induction of *Id3* and subsequent Id3-mediated disruption of E protein binding.

We designated the regions of overlapping HEB, E2A, and Egr2 binding as *Tcf12*^HE1^ and *Tcf12*^HE2^. Sequence-level analysis of *Tcf12*^HE1^ and *Tcf12*^HE2^ allowed the identification of predicted E protein binding sites (E box) in close proximity to Egr binding sites (Egr) (**Figure 7—figure supplement 1B**), suggesting that HEB/E2A and Egr2 may participate in a gene regulatory protein complex. Together, these findings support the hypothesis that HEB and E2A work in concert with Egr2 to modulate *Id3* transcription in thymocytes undergoing TCR-mediated selection.

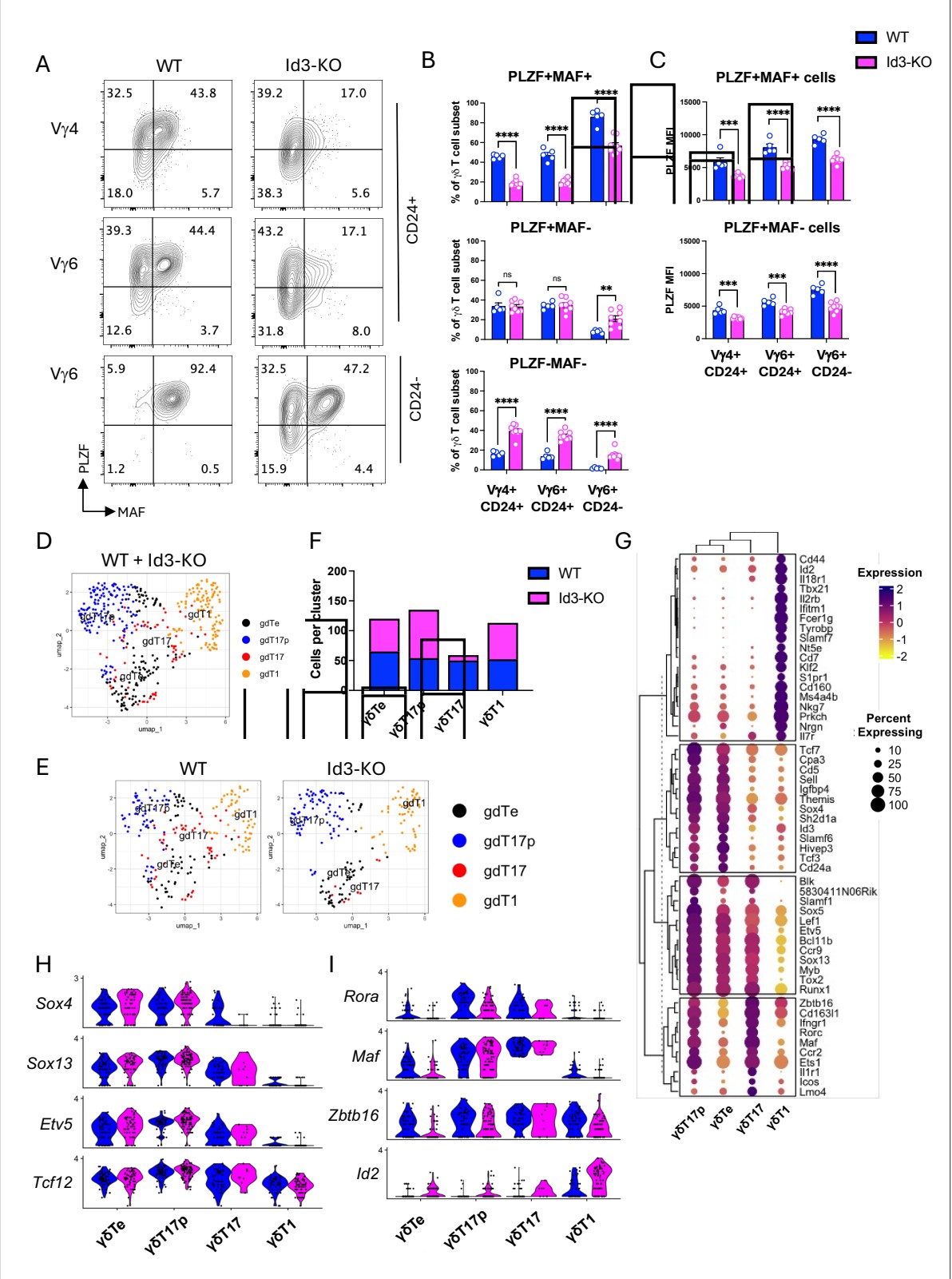

**Figure 6.** Intact γδ T commitment gene program and impaired γδT17 maturation program in Id3-KO mice. (**A**) Flow cytometry plots showing the percentages of cells expressing PLZF and/or MAF in immature (CD24⁺) Vγ4 and Vγ6 cells, and in mature (CD24⁻) Vγ6 cells (note that mature Vγ4 cells are not present in the embryonic day 18 [E18] fetal thymus). (**B**) Quantification of the percentages of cells expressing PLZF and MAF (top), PLZF only (middle), or neither (bottom) within the immature Vγ4 and Vγ6 subsets, and the mature Vγ6 subset. (**C**) Mean fluorescent intensities of PLZF in the

*Figure 6 continued*

PLZF$^+$MAF$^+$ and PLZF$^+$MAF$^-$ populations within the immature and mature Vγ subsets. (**D, E**) scRNA-seq UMAP plots of γδ T cells as merged (**D**) or split (**E**) into wild-type (WT) versus Id3-KO populations. (F) Number of WT and Id3-KO cells per cluster. (**G**) Clustered dot plot of curated gene sets used to assign γδT17p, γδTe, γδT17, and γδT1 identities. (**H**) Expression of γδ T cell commitment genes in WT versus Id3-KO cells by cluster. (**I**) Expression of γδT17 maturation genes in WT versus Id3-KO cells by cluster. Experiments were done two to three times, and results were pooled for analysis. Each biological replicate is depicted as an open circle on the bar graphs. Blue = WT, pink = Id3 KO. Significant differences between WT and Id3-KO subsets were determined using unpaired classic Student's t-tests. **p<0.01, ***p<0.001, ****p<0.0001.

The online version of this article includes the following figure supplement(s) for figure 6:

**Figure supplement 1.** Identification of γδ T cell subsets from embryonic day 18 (E18) wild-type (WT) and Id3-KO double negative (DN) cells using single-cell RNA sequencing (scRNA-seq).

**Figure supplement 2.** γδTCR$^+$ cells from embryonic day 18 (E18) Id3-KO mice include a population of CD4$^+$CD8$^+$ cells, indicating diversion to the αβ-T lineage program.

To examine how the chromatin landscape of the *Id3* locus might change across this transition, we generated ATAC-seq data from DN3 and DN4 cells, as well as *Rag2$^{-/-}$* DN3 cells, to provide a genuine pre-selection context (*Figure 7—figure supplement 1A*). Alignment of ATAC-seq and ChIP-seq peaks in the *Id3* locus revealed accessibility of *Tcf12$^{HE1}$* and *Tcf12$^{HE2}$* in *Rag2$^{-/-}$*, WT DN3, and WT DN4 cells, strengthening their relevance. Given the known ability of E2A and HEB to induce chromatin remodeling (*Emmanuel et al., 2018*; *Lin et al., 2010*), we also examined accessibility in DN3 and DN4 cells from HEB cKO mice (*Figure 7—figure supplement 1A*). Alignment of ATAC-seq and ChIP-seq peaks in the *Id3* locus revealed accessibility of *Tcf12$^{HE1}$* and *Tcf12$^{HE2}$* in *Rag2$^{-/-}$* DN3, WT DN3, and WT DN4 cells. However, accessibility of *Tcf12$^{HE1}$* and *Tcf12$^{HE2}$* was dampened in HEB cKO cells, especially at the DN3 stage, suggesting that HEB may be involved in remodeling the *Id3* locus, resulting in a poised state that enables TCR-dependent transcription factors like Egr2 to induce *Id3* proportionally to TCR signal strength.

## TCR signaling and HEB converge to potentiate *Id3* transcription

To test whether HEB can cooperate with TCR/CD3 signaling to upregulate *Id3*, we used a gain-of-function approach. We took advantage of the SCID.adh cell line, which expresses a chimeric hIL-2Rα:CD3ε receptor that mimics pre-TCR signaling when stimulated with anti-TAC (hIL-2Rα) antibody (*Carleton et al., 1999*). Stimulation results in downregulation of CD25, *Rag1*, *Rag2*, and *Ptcra*, while inducing *Trac* germline transcripts, recapitulating pre-TCR activity. We transduced SCID.adh cells with either control or HEB-expressing retroviruses to generate control and HEB-overexpressing cells (*Figure 7B*). To avoid the growth arrest and cell death associated with full-length HEB (HEBCan) overexpression (*Engel and Murre, 2004*; *Wang et al., 2010*), we used a construct encoding HEBAlt, a truncated form that activates E protein target genes without impairing cell viability (*Wang et al., 2010*; *Wang et al., 2006*; *Yoganathan et al., 2022*).

Upon overnight stimulation with anti-TAC, both control and HEB-expressing cells downregulated CD25 to similar degrees, as assessed by flow cytometry, confirming effective CD3-mediated signaling (*Figure 7C and E*). Quantitative RT-PCR (qRT-PCR) analysis showed that *Id3* expression was modestly elevated in unstimulated HEB-expressing cells compared to controls, and stimulation of control cells upregulated *Id3* levels to a greater degree (*Figure 7D*). Notably, *Id3* expression in stimulated cells expressing HEB was much higher than either HEB or stimulation alone, and exceeded the sum of these two conditions, indicating a synergistic interaction. These results demonstrate that HEB can amplify *Id3* induction in response to TCR signaling, potentially through cooperative interaction with TCR-induced factors such as Egr2.

## Discussion

HEB and Id3 are both critical for T cell development, but their distinct functions in fetal γδ T cell development had not been resolved. Here, we show that HEB controls expression of genes involved in γδTCR signaling and induces a transcriptional program that primes γδ T cell precursors for γδT17 differentiation. Furthermore, HEB collaborates with TCR-dependent factors to upregulate *Id3*, which enables γδT17 cell maturation and inhibits the αβ T cell fate. Together, these findings define a sequence

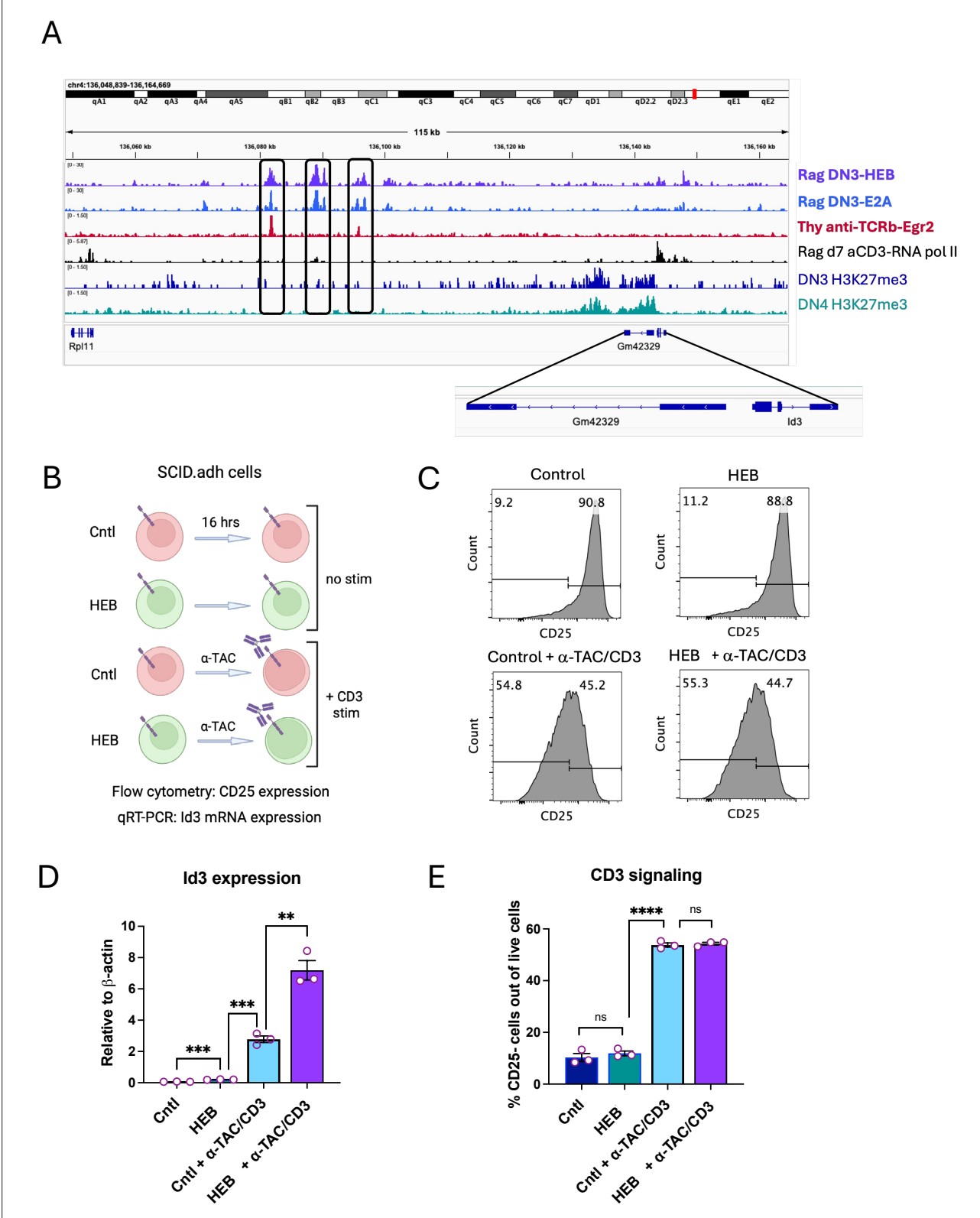

**Figure 7.** Synergistic upregulation of *Id3* by HEB and CD3 signaling. (**A**) ChIP-seq data analysis of the binding of HEB, E2A, RNA polymerase, and Egr2, and the extent of H3K27me3 chromatin modification, in DN3 and/or DN4 cells at the *Id3* gene locus, obtained from publicly available datasets (see Materials and methods for accession numbers). The cell type and antibody used in each experiment are indicated to the right of the tracks. Peaks bound by HEB, E2A, and/or Egr2 are indicated in boxes. Inset shows the *Id3* exons and the adjacent *Gm42329* long non-coding RNA. (**B**) Diagram

**eLife** Research article

Developmental Biology | Immunology and Inflammation

*Figure 7 continued*

of experimental design. SCID.adh cells transduced with HEBAlt or control retroviral vectors were cultured for 16 hr in the presence or absence of the anti-TAC antibody, which induces signaling through the CD3 complex. (**C, E**) Flow cytometry plots (**C**) and quantification (**E**) of CD25 upregulation with and without stimulation and/or HEB expression. (D) *Id3* mRNA expression relative to β-actin as determined by quantitative RT-PCR. Rag = *Rag2*[-/-] mouse thymocytes, which are arrested at the DN3 stage of development. Experiments were done two times, and results were pooled for analysis. Each biological replicate is depicted as an open circle on the bar graphs. Significant differences were determined using unpaired classic Student's t-tests. **p<0.01, ***p<0.001, ****p<0.0001.

The online version of this article includes the following figure supplement(s) for figure 7:

**Figure supplement 1.** Accessibility and occupancy of Id3 locus elements by HEB, E2A, and Egr2 before and after pre-T cell receptor (TCR) or γδTCR signaling.

**Figure supplement 2.** Model for HEB and Id3 requirements in the development and maturation of γδT17 cells.

of regulatory states governed by the E/Id axis that orchestrate γδ T cell lineage commitment and the differentiation of functional γδT17 cells.

TCR signal strength is tightly linked to specific combinations of Vγ and Vδ chains expressed on fetal γδ T cells, which serve as critical drivers of γδ T cell fate and functional programming (*Chen et al., 2021*; *Fahl et al., 2018b*; *Munoz-Ruiz et al., 2016*; *Scaramuzzino et al., 2022*). We found that HEB is required for the expression of TRGV4 and TRDV5. These genes encode the Vγ4 Vδ5 TCR, which supports γδT17 cell differentiation (*Kashani et al., 2015*). While TRGV4 is known to be regulated by E proteins (*Bain et al., 1999*; *In et al., 2017*; *Nozaki et al., 2011*), the requirement for HEB in TRDV5 regulation is newly appreciated. Notably, the TRD locus contains E2A-dependent insulators that limit TRDV4 expression to the fetal thymus (*Hao and Krangel, 2011*). Whether these are also HEB-dependent, and whether they impact TRDV expression among fetal γδ T cell subsets, remains to be determined.

Our analysis identified known E protein target genes, including components of the TCR signaling pathway (*Cd3d*, *Cd3g*, *Lat*, *Zap70*) (*Braunstein and Anderson, 2011*; *Miyazaki et al., 2017*) and chemokine receptors (*Cxcr5*, *Cxcr4*, *Ccr9*) (*Kadakia et al., 2019*; *Krishnamoorthy et al., 2015*; *Miyazaki et al., 2011*), validating the experimental approach. Moreover, we found that a suite of γδT17-specific genes were also HEB-dependent, including *Blk* and *Syk*, which are γδT17 cell-specific mediators of TCR signaling. We also noted that several inhibitors of TCR signaling were decreased, including *Pdcd1* (encodes PD-1), *Nfbia* (encodes Ikappaα), and *Sh2d2a* (encodes TSAd). All of these inhibitors are regulated post-translationally, suggesting a role for HEB in inducing T lineage-specific pathways that enable both positive and negative regulation of TCR signal transduction.

Our studies show that Id3 limits αβ lineage potential even in the presence of HEB, suggesting that HEB restricts the αβ program primarily by upregulating *Id3*. In contrast, the early γδT cell program is upregulated in Id3-KO mice, indicating that Id3 is not required for this function. A third E/Id dynamic operates during γδT17 maturation, which requires Id3 independently of HEB. Sox13 has been shown to initiate a cascade of regulatory events that induce *Blk* and *Maf* during γδT17 development (*In et al., 2017*; *Pokrovskii et al., 2020*). However, our work indicates that Sox13 is not sufficient to upregulate γδT17 maturation factors like *Zbtb16* and *Maf* in the absence of Id3, implying an E protein-dependent brake on γδT17 maturation. This highlights a decoupling between lineage specification and effector maturation, supporting the two-step model of γδ T cell development (*Buus et al., 2016*).

HEB and E2A can partially compensate for each other in the context of lymphocyte development, complicating interpretations of single gene knockouts (*Barndt et al., 2000*; *Zhuang et al., 1998*). This has been especially well characterized in αβ-T cell development, in which a lack of E2A leads to partial blocks at the ETP and DP to SP transitions (*Ikawa et al., 2006*; *Jones and Zhuang, 2007*), whereas HEB deficiency impairs the ISP to DP transition (*Barndt et al., 1999*). Mice conditionally lacking both HEB and E2A have more severe phenotypes (*Jones and Zhuang, 2011*), especially at the ETP to DN2 transition when T cell lineage identity is being established (*Miyazaki et al., 2017*), and at the DN2 to DN3 transition, when T lineage commitment occurs (*Wojciechowski et al., 2007*). Indeed, early loss of HEB alone renders T cell precursors more open to divergence to non-T lineages, even at the DN3 stage (*Braunstein and Anderson, 2011*). Our laboratory and others identified 'DN1-like' cells in the thymus of E protein-deficient and Id1-overexpressing mice, which were later identified as ILC2s (*Berrett et al., 2019*; *Miyazaki et al., 2025*; *Miyazaki et al., 2017*; *Qian et al., 2019*). The TCRγ rearrangements observed in thymic ILC2s and peripheral ILC2s in normal mice (*Pankow and Sun, 2022*;

*Qian et al., 2019*; *Shin et al., 2020*) support a critical role for E proteins in lineage divergence, rather than outgrowth of alternative lineages, even prior to the αβ/γδ T cell branch point.

Our findings are consistent with a selective requirement for HEB factors in γδT17 cell differentiation. Intriguingly, we observed that γδT1 lineage cells express *Tcf3* (E2A) and *Id3* but not *Tcf12* (HEB). Furthermore, *Tcf3* levels remained constant under conditions of HEB deficiency in all γδ T cell subsets. These results suggest that while E2A is insufficient for γδT17 development, it may be required and sufficient for γδT1 development. Overall, our analysis suggests a strong division of labor between HEB and E2A in regulating the γδT17 and γδT1 fates, respectively.

It is possible that Id3 is needed after γδ T cell specification to inhibit E2A-driven inhibitors of γδT17 maturation. Possible candidates include positive regulators of γδT1 differentiation, such as T-bet (*Tbx21*) or Eomes. These transcription factors participate in negative cross-regulatory loops with γδT17 regulators such as Runx1, RORγt, and AP-1 factors, which help to stabilize γδ T cell subset lineage identity (*Parker et al., 2025*). Additionally, transiently expressed HEB-dependent transcription factors such as *Etv5* and *Sox5* may act as a checkpoint for γδT17 maturation, to be released upon *Id3* upregulation.

The TCR signal strength model posits that strong signals induce high levels of Egr factors, which in turn induce high levels of Id3 (*Haks et al., 2005*; *Lauritsen et al., 2009*). Our data add an important new dimension to this paradigm, proposing that the lower Egr2 levels induced by weaker γδTCR signaling require cooperation with HEB/E2A to upregulate *Id3* expression (*Figure 7—figure supplement 2A*). This model is supported by the ChIP-seq data, which confirms HEB and E2A binding at the *Id3* locus, at regions also bound by Egr2, and gain-of-function studies, which show that HEB synergizes with CD3 signals to amplify *Id3* expression. However, it is important to note that additional studies will be needed to confirm whether HEB and E2A collaborate with Egr2 in a temporally coordinated fashion during γδ T cell specification. Both HEB and E2A are pioneering factors and can increase locus accessibility of genes involved in lymphoid development, such as *Foxo1* (*Welinder et al., 2011*). Interestingly, we found that HEB/E2A peaks in the *Id3* locus are diminished in cells from HEB cKO mice relative to WT, particularly at the DN3 stage. Therefore, HEB may play an important role in epigenetically priming the *Id3* locus prior to TCR signaling.

Our data is consistent with a partial compensation for Id3 by Id2, less effectively during γδ T cell commitment, and more fully during late γδT17 maturation, when *Id2* is normally expressed. This is consistent with the observation that Id3 supports transient developmental transitions, while Id2 stabilizes innate-like transcriptional states (*Anderson, 2022*). Since *Id2* is a direct target of E2A (*Schwartz et al., 2006*), the upregulation of *Id2* in *Id3*-deficient precursors likely reflects increased E protein activity. Thus, HEB, Id3, and Id2 participate in negative regulatory loops that allow transient HEB activity and *Id3* expression, followed by stabilization of the innate γδT17 gene network by Id2 (*Figure 7—figure supplement 2B*). Understanding how Id3 and Id2 differentially regulate the timing and magnitude of E protein activity and whether they have different impacts on E2A/E2A homodimers versus HEB/E2A heterodimers remains to be addressed.

There are limitations to our study. Our analyses focused on E18 γδ T cells, which reflect γδT17-biased fetal development and may not capture functions of HEB or Id3 at other stages. We previously showed that HEB cKO mice have defects in the production of functional γδT17 cells in neonatal thymus (*In et al., 2017*). We also found that γδ T cells from HEB cKO mice exhibited a diminished capacity for IL-17 production in adult lungs and spleen γδ T cells. While the adult thymus does not support the development of fully functional innate γδ T cells (*Chen et al., 2021*; *Haas et al., 2012*; *Kashani et al., 2015*), it does contain γδTCR+ cells with activated *Sox-Maf-Rorc* gene networks (*Yang et al., 2023*). In addition, we have not yet addressed the roles of HEB and Id3 in TCR-independent development of a subset of early fetal-derived Vγ4 γδ T cells (*Spidale et al., 2018*). Importantly, the specific contributions of HEBAlt and HEBCan isoforms remain unresolved, as both were deleted in our conditional HEB model. Further studies of γδ T cell development in the neonatal and adult thymus of HEB cKO, as well as HEBAlt KO and HEBCan KO mice, are underway.

In summary, our studies have identified multiple interlinked transcriptional circuits that require E proteins and Id factors during γδ T cell development. HEB induces Id3, which then inhibits E protein activity. In the absence of Id3, HEB and E2A activity persist, inducing compensatory Id2 expression. Since *Id3* expression is self-limiting through E protein suppression, the HEB-Id3 interactions result in a negative feedback loop. Thus, HEB plays dual roles in establishing γδ lineage identity and initiating

γδT17 differentiation via Id3. Future work should clarify the direct transcriptional targets and co-factors of HEB, and how dynamic levels of HEB, Id3, and Id2 coordinate γδ T cell fate decisions.

# Materials and methods

### Key resources table

| Reagent type (species) or resource | Designation | Source or reference | Identifiers | Additional information |
|---|---|---|---|---|
| Gene (*Mus musculus*) | *Tcf12* | GenBank | GenBank:NM_011544.3 | Encodes HEB isoforms HEBAlt and HEBCan |
| Gene (*Mus musculus*) | *Id3* | GenBank | GenBank:NM_008321.3 | Encodes Id3 |
| Strain, strain background (*Mus musculus*, C57BL/6, male and female) | *Tcf12*<sup>fl/fl</sup> (HEB<sup>fl/fl</sup>) | PMID:17442955 | Not commercially available, provided upon request | Conditional *Tcf12* floxed allele; bred to Vav-iCre to generate HEB cKO mice |
| Strain, strain background (*Mus musculus*, C57BL/6, male and female) | Vav-iCre | The Jackson Laboratory | IMSR:JAX:008610; RRID:IMSR_JAX:008610 | B6.Cg-Commd10Tg(Vav1-icre)A2Kio/J; hematopoietic Cre driver |
| Strain, strain background (*Mus musculus*, C57BL/6, male and female) | *Id3*-KO (*Id3*-RFP) | The Jackson Laboratory | IMSR:JAX:010983; RRID:IMSR_JAX:010983 | B6;129S-Id3tm1Pzg/J; backcrossed to C57BL/6 for 8 generations |
| Strain, strain background (*Mus musculus*, C57BL/6, male and female) | *Rag2*-KO | The Jackson Laboratory | IMSR:JAX:008449; RRID:IMSR_JAX:008449 | B6.Cg-Rag2tm1.1Cgn/J; lacks mature T and B cells |
| Cell line (*Mus musculus*) | SCID.adh | PMID:10452996 | Not commercially available, provided upon request | Pro-T cell line derived from SCID mice expressing a hybrid hIL2-CD3e signaling molecule; verified by flow cytometry (CD44+/–, CD25+) and downregulation of CD25 upon stimulation |
| Recombinant DNA reagent | MIGR1-HEBAlt retroviral vector | PMID:20826759 | Not commercially available, provided upon request | MSCV retroviral vector encoding HEBAlt downstream of an IRES-GFP; used to transduce SCID.adh cells; HEBAlt corresponds to Tcf12 transcript variant 4 (GenBank:NM_001253864.1) |
| Antibody | PE anti-CD4 (rat monoclonal, clone GK1.5) | Thermo Fisher Scientific | Cat#:12-0041-82; RRID:AB_465506 | (FACS, 1:200) |
| Antibody | FITC anti-mouse CD8a (rat monoclonal, clone 53–6.7) | BioLegend | Cat# 100705; RRID:AB_312744 | (FACS, 1:200) |
| Antibody | BUV737 anti-CD4 (rat monoclonal, clone GK1.5) | Thermo Fisher Scientific | Cat#:367-0041-82; RRID:AB_2895921 | (FACS, 1:200) |
| Antibody | APC-eFluor 780 anti-mouse CD8a (rat monoclonal, clone 53-6.7) | Thermo Fisher Scientific | Cat#:47-0081-82; RRID:AB_1272185 | (FACS, 1:200) |
| Antibody | PE-Cy7 anti-mouse CD25 (rat monoclonal, clone PC61.5) | BD Biosciences | Cat#:551071; RRID:AB_394031 | (FACS, 1:200) |
| Antibody | Alexa Fluor 700 anti-mouse CD3e (Armenian hamster monoclonal, clone 145-2C11) | BioLegend | Cat#:100236; RRID:AB_2561455 | (FACS, 1:200) |

*Continued on next page*

*Continued*

| Reagent type (species) or resource | Designation | Source or reference | Identifiers | Additional information |
|---|---|---|---|---|
| Antibody | BV605 anti-mouse CD73 (rat monoclonal, clone TY/11.8) | BD Biosciences | Cat#:752734 | (FACS, 1:200) |
| Antibody | BUV496 anti-mouse CD24 (rat monoclonal, clone M1/69) | BD Biosciences | Cat#:612953 | (FACS, 1:200) |
| Antibody | BUV737 anti-mouse CD27 (hamster monoclonal, clone LG.7F9) | BD Biosciences | Cat#: 612831 | (FACS, 1:200) |
| Antibody | PE anti-mouse PLZF (mouse monoclonal, clone R17-809) | BD Biosciences | Cat#:564850 | (FACS, 1:200) |
| Antibody | eFluor 450 anti-mouse c-Maf (mouse monoclonal, clone sym0F1) | Thermo Fisher Scientific | Cat#:48-9855-42; RRID:AB_2762608 | (FACS, 1:200) |
| Antibody | APC anti-mouse IL-17A (rat monoclonal, clone eBio17B7) | Thermo Fisher Scientific | Cat#:17-7177-81 | (FACS, 1:200) |
| Antibody | PerCP-eFluor 710 anti-mouse TCRgd (Armenian hamster monoclonal, clone GL3) | Thermo Fisher Scientific | Cat#:46-5711-82; RRID:AB_2016707 | (FACS, 1:200) |
| Antibody | BV421 anti-mouse TCRgd (hamster monoclonal, clone GL3) | BD Biosciences | Cat#:562892 | (FACS, 1:200) |
| Antibody | BV711 anti-mouse Vgamma1.1 TCR (Armenian hamster monoclonal, clone 2.11) | BD Biosciences | Cat#:745456 | (FACS, 1:200) |
| Antibody | PE anti-mouse Vgamma3 (rat monoclonal, clone 536) | BioLegend | Cat#:137504; RRID:AB_2562450 | (FACS, 1:200); commercial anti-Vgamma3 is referred to as Vgamma5 in the manuscript |
| Antibody | FITC anti-mouse Vgamma3 (rat monoclonal, clone 536) | BD Biosciences | Cat#:553229; RRID:AB_394747 | (FACS, 1:200); commercial anti-Vgamma3 is referred to as Vgamma5 in the manuscript |
| Antibody | PE-Cy7 anti-mouse Vgamma2 (Armenian hamster monoclonal, clone UC3-10A6) | Thermo Fisher Scientific | Cat#:25-5828-82; RRID:AB_2573474 | (FACS, 1:200); commercial anti-Vgamma2 is referred to as Vgamma4 in the manuscript |
| Antibody | APC anti-mouse Vgamma2 (Armenian hamster monoclonal, clone UC3-10A6) | BioLegend | Cat#:137707; RRID:AB_2563942 | (FACS, 1:200); commercial anti-Vgamma2 is referred to as Vgamma4 in the manuscript |
| Antibody | Anti-TAC/anti-human IL-2R alpha (mouse monoclonal) | PMID:33535043 | Not commercially available, provided by request | Plate-bound at 5 µg/mL; binding to human IL-2Ra on SCID.adh cells mimics pre-TCR signaling |
| Antibody | Anti-E2A (rabbit polyclonal) | PMID:33535043 | Other | (ChIP-seq, 10 µg/IP); affinity-purified rabbit polyclonal sera raised against the last 12 amino acids of the E2A C-terminus; see previously described ChIP-seq methods |
| Antibody | Anti-HEB (rabbit polyclonal) | PMID:33535043 | Other | (ChIP-seq, 10 µg/IP); affinity-purified rabbit polyclonal sera raised against the last 12 amino acids of the HEB C-terminus; see previously described ChIP-seq methods |

*Continued*

| Reagent type (species) or resource | Designation | Source or reference | Identifiers | Additional information |
|---|---|---|---|---|
| Sequence-based reagent | Id3 qRT-PCR primer set | Integrated DNA Technologies | Other | Forward: CTGTCGGAACGTAGCCTGG; Reverse: GTGGTTCATGTCGTCCAAGAG |
| Sequence-based reagent | Actb (beta-actin) qRT-PCR primer set | Integrated DNA Technologies | Other | Forward: ATGGTGGGAATGGGTCAGAA; Reverse: TCTCCATGTCGTCCCAGTTG |
| Commercial assay or kit | LIVE/DEAD Fixable Aqua Dead Cell Stain Kit, for 405 nm excitation | Thermo Fisher Scientific | Cat#:L34957 | Flow cytometry |
| Commercial assay or kit | SuperScript III First-Strand Synthesis System | Invitrogen | Cat#:18080051 | cDNA synthesis for qRT-PCR |
| Commercial assay or kit | Chromium Next GEM Single Cell 5' Kit v2 (Dual Index) | 10x Genomics | Cat#:PN-1000263 | scRNA-seq |
| Commercial assay or kit | Chromium Next GEM Single Cell 3' Kit v2 (Dual Index) | 10x Genomics | Cat#:PN-1000268 | scRNA-seq |
| Commercial assay or kit | PowerUp SYBR Green Master Mix | Thermo Fisher Scientific | Cat#:A25742 | qRT-PCR |
| Commercial assay or kit | FIX & PERM Cell Permeabilization Kit | Thermo Fisher Scientific | Cat#:88-8824-00 | Catalog number reported in supplied file |
| Commercial assay or kit | Foxp3/Transcription Factor Staining Buffer Set | Thermo Fisher Scientific | Cat#:00-5523-00 | Intracellular flow cytometry |
| Chemical compound, drug | TRIzol Reagent | Invitrogen | Cat#:15596026 | RNA extraction |
| Chemical compound, drug | Brefeldin A Solution (1000×) | Thermo Fisher Scientific | Cat#:00-4506-51 | Inhibition of cytokine secretion |
| Software, algorithm | BD FACSDiva Software | BD Biosciences | RRID:SCR_001456 | Flow cytometry acquisition and analysis |
| Software, algorithm | FlowJo | FlowJo, LLC | RRID:SCR_008520 | Flow cytometry analysis |
| Software, algorithm | STAR (v2.5.2b) | Dobin et al.; SciCrunch | RRID:SCR_004463 | FASTQ alignment to the mm39 genome |
| Software, algorithm | SAMTOOLS (v0.1.19) | HTSlib/SciCrunch | RRID:SCR_002105 | BAM file processing |
| Software, algorithm | BEDTools (v2.25.0) | bedtools/SciCrunch | RRID:SCR_006646 | BED file processing |
| Software, algorithm | Cell Ranger (v1.1.7) | 10x Genomics | RRID:SCR_017344 | Read alignment and matrix generation |
| Software, algorithm | Seurat (v4.4) | *Hao et al., 2021*; Satija Lab | RRID:SCR_016341 | Single-cell RNA-seq analysis; R markdown files used for downstream analysis |
| Software, algorithm | KEGG | Kanehisa Laboratories; SciCrunch | RRID:SCR_012773 | Pathway analysis |
| Software, algorithm | Cistrome Data Browser | Cistrome | RRID:SCR_000242 | Retrieval of public ChIP-seq and ATAC-seq datasets |
| Software, algorithm | Integrative Genomics Viewer (IGV) | Broad Institute | RRID:SCR_011793 | Visualization of genome-aligned sequencing data |
| Software, algorithm | ShinyGO | ShinyGO | RRID:SCR_019213 | Gene ontology analysis |

Continued

| Reagent type (species) or resource | Designation | Source or reference | Identifiers | Additional information |
|---|---|---|---|---|
| Software, algorithm | Immunological Genome Project (ImmGen) | ImmGen | RRID:SCR_021792 | Immune cell gene expression database |
| Other | Anti-CD4 MicroBeads | Miltenyi Biotec | Cat#:130-117-043 | MACS enrichment |
| Other | Anti-CD8a MicroBeads | Miltenyi Biotec | Cat#:130-117-044 | MACS enrichment |

## Experimental design and statistical analysis

The overall goal of this study was to understand how HEB transcription factors regulate the transcriptional networks required for the development of IL-17-producing γδ T cells. The subjects of these studies were genetically modified mice. The experimental variable was the genotype of the mice. For a given genotype, experimental mice were chosen randomly based on the availability of animals and littermate controls. Mice of both sexes were used. No sex differences were apparent; therefore, data were pooled. Experiments were done two to three times, depending on the number of mice available for each experiment, and results were pooled for statistical analysis for a total n of ≥3. Each biological replicate (mouse) is depicted as an open circle on the bar graphs. The investigators were not blinded to allocation during experiments and outcome assessment, except when fetal thymocytes were analyzed by flow cytometry prior to genotyping. Groups were unpaired, with similar variances, and all comparisons between genotypes were made within a given subset. Therefore, we assessed the significance of each comparison of mean values using an unpaired two-tailed classic Student's t-test, with $p < 0.05$ considered statistically significant. Error bars represent standard error of the mean (SEM) values.

## Mice

Genetically modified mice were used in this study. $Tcf12^{fl/fl}$ mice have loxP sites flanking the helix-loop-helix dimerization domain, which is shared by all HEB isoforms (**Wang et al., 2006**; **Wojciechowski et al., 2007**). These mice were bred to *Vav-iCre* transgenic mice (JAX stock #008610) (**de Boer et al., 2003**), which deletes regions flanked by loxP sites in all hematopoietic cells (**Siegemund et al., 2015**), to generate $Tcf12^{fl/fl}$ *Vav-iCre* (HEB cKO) mice, as previously described (**In et al., 2017**; **Welinder et al., 2011**). All experimental HEB cKO mice were obtained from timed matings of $Tcf12^{fl/fl}$ x $Tcf12^{fl/fl}$ *Vav-iCre* mice, enabling parallel analysis with WT (no Cre) littermates. Id3-deficient mice (Id3-KO) lacked *Id3* in all cells due to a knock-in/knockout allele in which the *Id3* coding sequence was replaced with red fluorescent proteins (RFP) (JAX strain # 010983, B6;129S-$Id3^{tm1Pzg}$/J). RFP was not detected in our analyses due to quenching during intracellular staining. Id3-KO mice were acquired on the E29/B6 background and bred onto the C57Bl/6 background for 8 generations. Id3$^{+/-}$ mice were timed mated together to produce WT and Id3-KO littermates. $Rag2^{-/-}$ mice were obtained from Jackson Labs (JAX strain # 008449, B6.Cg-Rag2tm1.1Cgn/J). Fetal thymocytes were analyzed at E18, and adult mice were analyzed at 6–9 weeks of age. Mice were bred and maintained in the Comparative Research Facility of the Sunnybrook Research Institute (Toronto, ON, Canada) under specific pathogen-free conditions. All animal procedures were approved by the Sunnybrook Research Institute Animal Care Committee under animal user protocol (AUP) 22204 and in compliance with the Canadian Council of Animal Care guidelines.

## Timed matings and embryo harvest

Timed matings were performed by setting up mating pairs (day 0) and separating them ~16 hr later. Embryos were harvested after 18 days (E18). Fetal thymuses were dissected and individually pressed through a 40 μm mesh with a syringe plunger to create single-cell suspensions. Cells were resuspended in 1× HBSS/BSA for sorting or flow cytometry. Embryos were processed and analyzed separately, and tail tissue was collected for genotyping. Genotyping was performed as previously described for HEB cKO mice (**Wojciechowski et al., 2007**). Id3-KO mice were genotyped using the protocol described for the B6;129S-$Id3^{tm1Pzg}$/J strain on the Jackson Laboratories website.

## Flow cytometry

Antibodies were purchased from eBiosciences (San Diego, CA, USA), BioLegend (San Diego, CA, USA), and BD Biosciences (Mississauga, ON, Canada). For flow cytometry, cells were washed and incubated

with Fc blocking antibody (BD Biosciences), followed by extracellular staining for surface CD4 (clone GK1.5), CD8α (clone 53–6.7), CD3 (clone 145-2C11), TCRγδ (clone GL3), Vγ4 (Vγ2; clone UC3-10A6), Vγ5 (Vγ3; clone 536), Vγ1 (Vγ1.1; clone 2.11), CD27 (clone LG3-1A10), CD24 (clone M1/69), CD73 (clone TY11.8), and CD25 (clone PC615). To assess functional capacities, cells were stimulated by incubation for 4 hr with PMA (50 ng/ml) and ionomycin (500 ng/ml) in the presence of Brefeldin A (5 mg/ml; eBioscience), washed with 1× HBSS/BSA, and incubated with Fc block before staining for surface epitopes. Cells were then fixed and permeabilized (Fix and Perm Cell Permeabilization Kit; eBioscience) and stained with antibodies against IL-17A (clone eBio17B7). For intracellular transcription factor staining, the cells were fixed, permeabilized (FoxP3 Staining Kit, eBioscience), and stained for MAF (clone sym0F1) and PLZF (clone R17-809). All flow cytometric analyses were performed using Becton-Dickinson (BD) LSRII, Fortessa, or SymphonyA5-SE cytometers. FACSDiva and FlowJo software were used for analysis. Sorting was performed on BD ARIA and BD Fusion sorters.

## Cell culture

SCID.adh cells, which have been engineered to express a surface human IL-2Ra (TAC):CD3epsilon chimeric signaling molecule on the surface, were cultured as previously described (*Anderson et al., 2002*). SCID.adh cell line identity was confirmed by flow cytometry phenotyping of CD44 and CD25 expression and response to stimulation with downregulation of CD25, and the cells tested negative for mycoplasma. SCID.adh cells were transduced with MIGR1 encoding GFP only (control) or MIGR1 encoding GFP and HEBAlt. GFP+ cells were sorted, expanded, and cultured overnight on plates coated with anti-TAC antibody at 5 µg/ml in 500 µl of PBS or PBS only as control. Cells were analyzed by flow cytometry for expression of CD25 and for mRNA expression of *Id3* by qRT-PCR.

## RNA extraction and qRT-PCR

Total RNA was extracted from cells using TRIzol Reagent (Invitrogen) and reverse-transcribed into complementary DNA using Superscript III (Invitrogen). Reactions for qRT-PCR were prepared using PowerUp SYBR Green Master Mix (Thermo Fisher) and 0.5 µM of primers. Primers (5′ to 3′) were as follows: β-actin forward: ATGGTGGGAATGGGTCAGAA, β-actin reverse: TCTCCATGTCGTCCCAGTTG, Id3 forward: CTGTCGGAACGTAGCCTGG, Id3 reverse: GTGGTTCATGTCGTCCAAGAG. The qRT-PCR was run and analyzed using an Applied Biosystems 7500 Fast Real-Time PCR System (Thermo Fisher) and a QuantStudio 5 Real-Time PCR System (Thermo Fisher). Values from the qRT-PCRs were normalized to β-actin values, and relative expression values were calculated by the delta Ct method.

## scRNA-seq of WT and HEB cKO γδ T cells

E18 fetal thymocytes from HEB cKO mice were incubated with Fc block for 30 min on ice. Small aliquots from each sample were stained with CD4, CD8, and CD3 to assess genotypes by flow cytometry, with a lack of DP indicating HEB deletion, and genotypes were verified by PCR. Five WT and five HEB cKO littermates were pooled by genotype, stained, and sorted for TCRγδ+CD3+ cells. Samples were not hash-tagged as this technology was not available at the time of the experiment. Fifty thousand cells per sample were loaded onto the 10× Chromium controller, and barcoded libraries were generated using the Chromium Next GEM Single Cell 5′ Kit v2 (Dual Index) (10x Genomics) at the Princess Margaret Genomics Facility (Toronto, ON, Canada). Next-generation Illumina sequencing was performed to a depth of ~30,000 reads per cell. The estimated number of cells sequenced was ~3300 for WT and ~5500 for HEB cKO samples.

## scRNA-seq of WT and Id3-KO DN fetal thymocytes

E18 fetal thymuses from Id3-KO mice were dissected from embryos, pressed through mesh as above, and incubated with Fc block. Embryos were genotyped by PCR, and thymocytes were pooled according to genotype (3 WT and 3 Id3-KO mice). DN cells were enriched by magnetic sorting using anti-CD4 and anti-CD8 microbeads according to the manufacturer's instructions (Miltenyi Biotech). Flow-through (CD4-CD8-) cells were processed in-house using the Chromium Next GEM Single Cell 3′ Kit v2 (Dual Index) (10x Genomics). Next-generation sequencing was performed using the Illumina platform to a depth of ~100,000 reads per cell. The estimated number of cells sequenced for each sample was ~3700.

## scRNA-seq data analysis of WT and HEB cKO $\gamma\delta$ T cells

FASTQ raw data files were aligned to the mm10 genome using the STAR aligner (STAR v2.5.2b). Cell Ranger (v1.1.7; 10x Genomics) (*Zheng et al., 2017*) was used to generate matrix files, which were analyzed using programs in R-Seurat version 4.4 (*Hao et al., 2021*), as detailed in the R-Markdown files. WT and HEB cKO datasets were processed for quality control by excluding cells with more than 7% mitochondrial genes, less than 1000 unique genes, and/or less than 4000 transcripts, resulting in 1272 WT cells and 1951 HEB cKO cells for further analysis. Filtered WT and HEB cKO datasets were merged, and SCTransform was applied to the merged dataset. Cell cycle regression was performed to mitigate the influence of cell cycle heterogeneity on clustering, and cells expressing high levels of *Lyz2* were excluded to remove most myeloid cells. PCA was performed (RunPCA) and used to compute a nearest neighbor graph (FindNeighbors) and identify clusters (FindClusters). UMAP plots were generated using RunUMAP and displayed using DimPlot_scCustom. FindMarkers was used to identify the top 10 most differentially expressed genes between clusters.

## Generation of gene lists and module scores

To construct an unbiased and comprehensive list of genes diagnostic for different stages and lineages of $\gamma\delta$ T cell development, we collected lists of differentially expressed genes from eight publications characterizing $\gamma\delta$ T cell subsets in the fetal thymus, adult thymus, and adult peripheral tissues (*du Halgouet et al., 2024*; *Inácio et al., 2025*; *Liu et al., 2020*; *Mistri et al., 2024*; *Narayan et al., 2012*; *Pokrovskii et al., 2020*; *Spidale et al., 2018*; *Yang et al., 2023*). Genes were filtered to remove those involved in cell cycle and metabolism, as well as NK cell receptor genes. Lists were combined, duplicates were removed, and a final list of 87 genes was obtained (*Figure 2—source data 1*). This list was used for computation of the top 10 genes per cluster that were most differentially expressed from all other clusters, visualized as a Clustered DotPlot. Comparisons with the source literature and the Immunological Genome Project (*Heng et al., 2008*) were used to categorize the clusters. $\gamma\delta$Te1 and $\gamma\delta$Te2 represent two types of early $\gamma\delta$ T cell subsets with random assignments into '1' and '2'.

As developmental stages are continuous, not discrete, some genes were present in more than one module; scores were based on all genes in each module. Genes in each module are listed below.

> gdTe1: *Ccr9, Sox13, Cpa3, Etv5, Sox5, Tox2, Ccr2, Igfbp4, Blk, Lmo4, Slamf1, Ifngr1, Rorc, Maf, Icos*
> gdTe2: *Hivep3, Cd28, Themis, Slamf6, Sell, Igfbp4, Cd24a, Gzma*
> gdT17p: *Blk, Il17re, Vdr, Il17a, Lmo4, Slamf1, Ifngr1, Rorc, Maf, Icos*
> gdT17: *Il18r1, Il7r, Id2, Il23r, Il1r1, Cd44*
> gdT1p: *Klf2, Nrgn, Cd2, Ms4a4b, Prkch, Nr4a1, Id3*
> gdT1: *S1pr1, Eomes, Slamf7, Tyrobp, Ifitm1, Fcer1g, Tbx21, Gzmb*
> abT: *Cd8a, Notch1, Cd8b1, Rag1, Rag2, Rmnd5a*

Module location and intensity in WT versus HEB cKO cells were visualized using split UMAPs. Comparisons of the expression of single genes between WT and HEB cKO clusters were shown using split violin plots. To compare clusters 1 and 4, we performed FindMarkers, using the Wilcox test. Differentially expressed genes were displayed using an Enhanced Volcano Plot, with significance set at $\log_2$fold change>0.5 and $-\log_{10}$p<$10^{26}$.

## scRNA-seq data analysis of WT and Id3-KO DN thymocytes

The following analyses were conducted on the Id3-KO dataset. FASTQ raw data files were aligned to the mm39 genome using the STAR aligner (STAR v2.5.2b), and Cell Ranger was used to generate matrix files, which were analyzed using programs in R-Seurat. Cells were computationally filtered as described above in the HEB cKO dataset, resulting in 2323 WT cells and 2893 Id3-KO cells for further analysis. Datasets were merged and subjected to SCTransform as above. PCA plots were generated and used to construct UMAP plots depicting clusters in merged datasets and distribution of WT versus Id3-KO cells within each cluster by UMAP (see R-markdown file). Feature plots and violin plots were used to identify clusters with $\gamma\delta$ T cell characteristics, which were subsetted using FindClusters. ClusteredDotPlot was used to visualize the top eight most differentially expressed genes within the curated gene set described above. Split violin plots were generated to show relative expression of early and late $\gamma\delta$ T cell genes in WT versus HEB cKO cells within each cluster.

## ATAC-seq

Thymuses were dissected from adult *Rag2*[-/-], WT, and HEB cKO mice, pressed through mesh as above, and incubated with Fc block. WT and HEB cKO DN cells were enriched by magnetic sorting using anti-CD4 and anti-CD8 microbeads according to the manufacturer's instructions (Miltenyi Biotech). Flow-through (CD4[-]CD8[-]) cells were stained and flow-sorted into two populations: DN3 (CD4[-]CD8[-]CD44[-]CD25[+]) cells and DN4 (CD4[-]CD8[-]CD44[-]CD25[-]). *Rag2*[-/-] DN3 cells were obtained from *Rag2*[-/-] mice, which do not develop past the DN3 stage, restricting the *Rag2*[-/-] DN3 population to pre-selection cells (*Shinkai et al., 1992*). Duplicates were generated for each subset, with each biological replicate derived from three mice. Both males and females were used. Sorted cells were cryopreserved in 50% FBS/40% growth media/10% DMSO in aliquots of 100,000 cells, which were subjected to ATAC-seq, as follows. After membrane permeabilization, a transposase loaded with sequencing adaptors was added, which mediated insertion of adaptors at accessible genomic locations. Libraries were amplified and subjected to paired-end next-generation sequencing using the Illumina platform. Paired-end ATAC-seq libraries were generated and sequenced by Active Motif (Carlsbad, CA, USA).

## ATAC-seq analysis

Bcl2fastq2 (v2.20) was used to process Illumina base-call data and perform demultiplexing, and bwa (v0.7.12) was used to align reads to the reference genome (mm10). Samtools (v0.1.19) was used to process BAM files. BEDtools (v2.25.0) was used to process BED files, and wigToBigWig (v4) was used to generate bigWIG files. bigWIG files were aligned using the Integrative Genomics Viewer (IgV) software (v2.16.2). Each sample yielded ~65 million mapped reads.

## Gene ontology analysis

Genes differentially expressed between cluster 1 and cluster 4 with a significance of $\log_2$FC>0.25 and adjusted p-value<0.001 (*Figure 4—source data 1*) were submitted to ShinyGO 8.0 (*Ge et al., 2020*) for gene ontology analysis. Pathway inclusion was set at a minimum of five genes with a false discovery rate (FDR) of 0.05. Genes were analyzed using the KEGG pathway database (*Kanehisa et al., 2025*; *Figure 4—source data 1*). Results were visualized as a bar graph showing fold enrichment (numbers) and $-\log_{10}$(FDR) values (colors).

## Alignment of ChIP-seq data

ChIP-seq data for *Rag2*[-/-] DN3 and Rag DN3 + γδTCR cells binding to HEB and E2A were generated as follows: *Rag2*[-/-] fetal liver hematopoietic cells were cultured on OP9-DL4 stroma to produce DN3 cells. *Rag2*[-/-] DN3 cells were retrovirally transduced with the KN6 γδTCR or control (no TCR), and sorted GFP[+] DN3 cells were cultured for 4 days on stroma expressing the weak KN6 ligand T10 to initiate γδTCR signaling. These cells were subjected to ChIP-seq using anti-E2A and anti-HEB antibodies as previously described (*Fahl et al., 2021*). All other files were obtained from the Cistrome database (*Zheng et al., 2019*). Peaks were aligned to the mouse genome (mm38) using the IgV (*Robinson et al., 2011*). Sources were as follows: Thy anti-TCRb-Egr2, GEO accession # GSM845900 (*Seiler et al., 2012*); Rag d7 aCD3-RNA pol II, GEO accession # GSM1340642 (*Pekowska et al., 2011*); DN3 H3K27me3, GEO accession # GSM1498423, and DN4 H3K27me3, GEO accession # GSM1498422 (*Oravecz et al., 2015*).

## Acknowledgements

We would like to thank Lisa Wells, Madeline Harvey, Vivien Musiime, and the SRI Animal Facility for excellent mouse care. We thank Yuan Zhuang (Duke University) for provision of the HEB cKO mice. We are indebted to the SickKids Flow Cytometry Core (supported by the Canadian Foundation for Innovation and the SickKids' Foundation) for antibody panel design and high-parameter flow cytometry analysis, and the SRI Flow Cytometry and Microscopy Core for flow cytometry and sorting. We also thank the UHN Princess Margaret Genomics Facility for construction of the WT and HEB cKO scRNA-seq libraries. The Digital Research Alliance of Canada (RRG #5070) provided cloud computing capacity for scRNA-seq data analysis.

## Additional information

### Funding

| Funder | Grant reference number | Author |
|---|---|---|
| National Institutes of Health | 1P01AI102853-06 | Juan Carlos Zúñiga-Pflücker |
| Canadian Institutes of Health Research | PJT153058 | Michele Kay Anderson |
| American Association of Immunologists | | Johanna S Selvaratnam |
| Canadian Institutes of Health Research | FDN154332 | Juan Carlos Zúñiga-Pflücker |
| Canadian Institutes of Health Research | PJT192050 | Juan Carlos Zúñiga-Pflücker |
| Canadian Institutes of Health Research | PJT165973 | Cynthia J Guidos |
| Canadian Institutes of Health Research | PPE196061 | Michele Kay Anderson |

The funders had no role in study design, data collection and interpretation, or the decision to submit the work for publication.

### Author contributions

Johanna S Selvaratnam, Conceptualization, Formal analysis, Validation, Investigation, Visualization, Methodology, Writing – review and editing; Juliana DB da Rocha, Formal analysis, Supervision, Validation, Investigation, Methodology, Writing – original draft; Vinothkumar Rajan, Conceptualization, Data curation, Software, Formal analysis, Investigation, Visualization, Methodology, Writing – review and editing; Helen Wang, Formal analysis, Validation, Visualization, Methodology, Writing – review and editing; Emily C Reddy, Formal analysis, Validation, Investigation, Methodology, Writing – review and editing; Miki S Gams, Investigation, Methodology, Writing – review and editing; Jenny Jiahuan Liu, Conceptualization, Formal analysis, Validation, Investigation, Methodology, Writing – review and editing; Cornelis Murre, David Wiest, Conceptualization, Resources, Funding acquisition, Methodology, Writing – review and editing; Cynthia J Guidos, Conceptualization, Formal analysis, Supervision, Funding acquisition, Methodology, Writing – review and editing; Juan Carlos Zúñiga-Pflücker, Conceptualization, Resources, Formal analysis, Supervision, Funding acquisition, Methodology, Writing – original draft, Project administration; Michele Kay Anderson, Conceptualization, Data curation, Formal analysis, Supervision, Funding acquisition, Investigation, Visualization, Writing – original draft, Project administration

### Author ORCIDs

Juan Carlos Zúñiga-Pflücker ⓘ https://orcid.org/0000-0003-2538-3178
Michele Kay Anderson ⓘ https://orcid.org/0000-0002-8820-5910

### Ethics

Mice were bred and maintained in the Comparative Research Facility of the Sunnybrook Research Institute (Toronto, Ontario, Canada) under specific pathogen-free conditions. All animal procedures were approved by the Sunnybrook Research Institute Animal Care Committee under animal user protocol (AUP) 22204 and in compliance with the Canadian Council of Animal Care guidelines.

Reviewer #1 (Public review): https://doi.org/10.7554/eLife.109197.3.sa1
Reviewer #2 (Public review): https://doi.org/10.7554/eLife.109197.3.sa2
Reviewer #3 (Public review): https://doi.org/10.7554/eLife.109197.3.sa3
Author response https://doi.org/10.7554/eLife.109197.3.sa4

# Additional files

## Supplementary files
MDAR checklist

## Data availability
Matrix files and R-markdown code for the HEB and Id3 scRNA-seq have been deposited to the Dryad repository and can be accessed within the "Single cell RNA-seq data of E18 fetal thymocytes from HEB Vav-iCre and Id3-KO mice and their wild type littermate counterparts" dataset: https://doi.org/10.5061/dryad.08kprr5cq. ATAC-seq bigwig files can be accessed within the Dryad repository in the "ATAC-seq data from DN3 and DN4 cells from WT, HEBcKO, and Rag2-KO mice" dataset: https://doi.org/10.5061/dryad.4tmpg4fr5.

The following datasets were generated:

| Author(s) | Year | Dataset title | Dataset URL | Database and Identifier |
|---|---|---|---|---|
| Selvaratnam JS, Dutra Barbosa da Rocha J, Rajan V, Wang H, Reddy EC, Liu JJ, Gams M, Murre C, Wiest D, Guidos CJ, Zuniga-Pflucker J, Anderson MK | 2025 | Data from: Single cell RNA-seq data of E18 fetal thymocytes from HEB Vav-iCre and Id3-KO mice and their wild type littermate counterparts | https://doi.org/10.5061/dryad.08kprr5cq | Dryad Digital Repository, 10.5061/dryad.08kprr5cq |
| Selvaratnam JS, da Rocha JD, Rajan V, Wang H, Reddy EC, Liu JJ, Gams MS, Murre C, Wiest D, Guidos CJ, Zúñiga-Pflücker J, Anderson MK | 2026 | ATAC-seq data from DN3 and DN4 cells from WT, HEBcKO, and Rag2-KO mice | https://doi.org/10.5061/dryad.4tmpg4fr5 | Dryad Digital Repository, 10.5061/dryad.4tmpg4fr5 |

The following previously published datasets were used:

| Author(s) | Year | Dataset title | Dataset URL | Database and Identifier |
|---|---|---|---|---|
| Fahl SP, Contreras AV, Verma A, Qiu X, Harly C, Radtke F, Zúñiga-Pflücker JC, Murre C, Xue HH, Sen JM, Wiest DL | 2021 | Chip-Seq analysis on developing gamma-delta T cells | https://www.ncbi.nlm.nih.gov/geo/query/acc.cgi?acc=GSM162290 | NCBI Gene Expression Omnibus, GSM162290 |
| Seiler MP, Mathew R, Liszewski MK, Spooner CJ, Barr K, Meng F, Singh H, Bendelac A | 2012 | Anti-TCRb_inj | https://www.ncbi.nlm.nih.gov/geo/query/acc.cgi?acc=GSM845900 | NCBI Gene Expression Omnibus, GSM845900 |
| Pekowska A, Benoukraf T, Zacarias-Cabeza J, Belhocine M, Koch F, Holota H, Imbert J, Andrau JC, Ferrier P, Spicuglia S | 2011 | PolII N20 RagCD3 | https://www.ncbi.nlm.nih.gov/geo/query/acc.cgi?acc=GSM1340642 | NCBI Gene Expression Omnibus, GSM1340642 |
| Oravecz A, Apostolov A, Polak K, Jost B, Le Gras S, Chan S, Kastner P | 2015 | WT_DN3_H3K27me3_exp2 | https://www.ncbi.nlm.nih.gov/geo/query/acc.cgi?acc=GSM1498422 | NCBI Gene Expression Omnibus, GSM1498422 |

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
