## [Editor Report · eLife Assessment]

The study provides **important** mechanistic insight into the transcriptional control of γδT17 development, elegantly demonstrating how HEB and Id3 act sequentially and cooperatively to regulate γδT17 cell specification and maturation. The study provides **compelling** evidence that advances the understanding of E-Id protein dynamics in thymic T cell specification. The work is comprehensive, technically rigorous, and conceptually clear, and will be of interest to immunologists, developmental biologists, and those studying the molecular underpinnings of physiological outcomes.

---

## [Referee Report · Reviewer #1 (Public review)]

The authors use Flow cytometry and scRNA seq to identify and characterize the defect in gdT17 cell development from HEB f/f, Vav-icre (HEB cKO) and Id3 germline-deficient mice. HEB cKO mice showed defects in the gdT17 program at an early stage, and failed to properly upregulate expression of Id3 along with other genes downstream of TCR signaling. Id3KO mice showed a later defect in maturation. The results together indicate HEB and Id3 act sequentially during gdT17 development. The authors further showed that HEB and TCR signaling synergize to upregulate Id3 expression in the Scid-adh DN3-like T cell line. Analysis of previously published Chip-seq data revealed binding of HEB (and Egr2) at overlapping regulatory regions near Id3 in DN3 cells.The study provides insight into mechanisms by which HEB and Id3 act to mediate gdT17 specification and maturation. The work is well performed and clearly presented.

Comments on revisions:

The authors have answered all of my questions. I am strongly supportive of the revised work.

---

## [Referee Report · Reviewer #2 (Public review)]

Summary:

The manuscript by Selvaratnam et al. defines how the transcription factor HEB integrates with TCR signaling to regulate Id3 expression in the context of gdT17 maturation in the fetal thymus. Using conditional HEB ablation driven by Vav Cre, flow cytometry, scRNA-seq, and reanalysis of ChIP-seq data the authors, provide evidence for a sequential model in which HEB and TCR-induced Egr2 cooperatively upregulate Id3, enabling gdT17 maturation and limiting diversion to the ab lineages. The work provides an important mechanistic insight into how the E/ID-protein axis coordinates gd T cell specification and effector maturation.

Strengths include:

(1) The proposed model that HEB primes, TCR induces, and Id3 stabilizes gdT17 cells in embryonal development is elegant and consistent with the findings.

(2) The choice of animal models and the study of a precise developmental window.

(3) The cross-validation of flow, scRNA-seq, and ChIP-seq reanalyses strengthens the conclusions.

(4) The study clarifies the dual role of Id3, first as an HEB-dependent maturation factor for gdT17 cells, and as a suppressor of diversion to the ab lineages.

Comments on revisions:

In this revised version of their manuscript the authors have effectively addressed all of my previous concerns. In its current form the study represents a significant advancement in our understanding of how the transcription factor HEB integrates with TCR signaling to regulate Id3 expression in the context of gdT17 maturation in the fetal thymus. In this revised version of their manuscript the authors have effectively addressed all of my previous concerns. In its current form the study represents a significant advancement in our understanding of how the transcription factor HEB integrates with TCR signaling to regulate Id3 expression in the context of gdT17 maturation in the fetal thymus.

---

## [Referee Report · Reviewer #3 (Public review)]

Summary:

The authors of this manuscript have addressed a key concept in T cell development: how early thymus gd T cells subsets are specified and the elements that govern gd T17 versus other gd T cell subset or ab T cell subsets are specified. They show that the transcriptional regulator HEB/Tcf12 plays a critical role in specifying the gd T17 lineage and, intriguingly that it up regulates the inhibitor Id3 which is later required for further gd T17 maturation.

Strengths:

The conclusions drawn by the authors are amply supported by a detailed analysis of various stages of T cell maturation in WT and KO mouse strains at the single cell level both phenotypically, by flow cytometry for various diagnostic surface markers, and transcriptionally, by single cell sequencing. Their conclusions are balanced and well supported by the data and citations of previous literature.

Weaknesses:

I actually found this work to be quite comprehensive.

Comments on revisions:

Nothing to add here. The authors were very thorough in their original submission, and all minor issues identified have been addressed to my satisfaction.

---

## [Author Response]

The following is the authors’ response to the original reviews.

We thank the reviewers for their enthusiasm and insightful suggestions. Our responses to specific concerns and questions are detailed below.

**Public Reviews**:
**Reviewer #1 (Public review):**
The authors use Flow cytometry and scRNA seq to identify and characterize the defect in gdT17 cell development from HEB f/f, Vav-icre (HEB cKO), and Id3 germline-deficient mice. HEB cKO mice showed defects in the gdT17 program at an early stage, and failed to properly upregulate expression of Id3 along with other genes downstream of TCR signaling. Id3KO mice showed a later defect in maturation. The results together indicate HEB and Id3 act sequentially during gdT17 development. The authors further showed that HEB and TCR signaling synergize to upregulate Id3 expression in the Scid-adh DN3-like T cell line. Analysis of previously published Chi-seq data revealed binding of HEB (and Egr2) at overlapping regulatory regions near Id3 in DN3 cells.The study provides insight into mechanisms by which HEB and Id3 act to mediate gdT17 specification and maturation. The work is well performed and clearly presented. We only have minor comments.
**Reviewer #2 (Public review):**
Summary:The manuscript by Selvaratnam et al. defines how the transcription factor HEB integrates with TCR signaling to regulate Id3 expression in the context of gdT17 maturation in the fetal thymus. Using conditional HEB ablation driven by Vav Cre, flow cytometry, scRNA-seq, and reanalysis of ChIP-seq data the authors, provide evidence for a sequential model in which HEB and TCR-induced Egr2 cooperatively upregulate Id3, enabling gdT17 maturation and limiting diversion to the ab lineages. The work provides an important mechanistic insight into how the E/ID-protein axis coordinates gd T cell specification and effector maturation.Strengths include:(1) The proposed model that HEB primes, TCR induces, and Id3 stabilizes gdT17 cells in embryonal development is elegant and consistent with the findings.(2) The choice of animal models and the study of a precise developmental window.(3) The cross-validation of flow, scRNA-seq, and ChIP-seq reanalyses strengthens the conclusions.(4) The study clarifies the dual role of Id3, first as an HEB-dependent maturation factor for gdT17 cells, and as a suppressor of diversion to the ab lineages.Weaknesses:(1) The ChIP-seq reanalysis indicates overlapping HEB, E2A, and Egr2 peaks ~60 kb upstream of Id3. Given that the Egr2 data are not generated using the same thymocyte subsets, some form of validation should be considered for the co-binding of HEB and Egr2, potentially ChIP-qPCR in sorted gdT17 progenitors.

We agree that this is a valid concern and continue to work on confirming the mechanism from several other angles. Validating HEB/E2A and Egr2 co-binding in gdT17 cell progenitors by ChIP-qPCR would/will be a very precise and definitive experiment, but it will be very challenging to perform, in part due to the low numbers of gdT17 precursors in the fetal thymus (note the y-axis scales in Fig. 1F, J). As a complementary approach, we have analyzed additional ChIP-seq data for HEB/E2A binding in Rag2^-/-^ DN3 cells retrovirally transduced with the KN6 gdTCR cultured with stroma expressing the weak KN6 ligand T10 for 4 days. This analysis revealed that the binding of HEB/E2A on those sites persisted after weak gdTCR signaling, strengthening the likelihood that concurrent binding of HEB/E2A and Egr2 occurs during this developmental transition. We noted that HEB/E2A binding was slightly dampened in Rag2^-/-^ DN3 + gdTCR cells relative to Rag2^-/-^ DN3 cells, consistent with the induction of Id3 and subsequent Id3-mediated disruption of E protein binding. We also located HEB/E2A and Egr binding sites in close proximity in the two regions that shared peaks between HEB/E2A and Egr2 analyses (HE1 and HE2), in line with the potential participation of these two transcription factors in an enhanceosome binding complex.

Furthermore, we examined the chromatin landscape of the Id3 locus by sorting WT DN3 and DN4 cells, as well as Rag2^-/-^ DN3 cells to provide a genuine pre-selection context, and performing ATAC-seq (Figure 7–suppl 7A). Given the known ability of E2A and HEB to induce chromatin remodeling, we also examined accessibility in DN3 and DN4 cells from HEB cKO mice. Alignment of ATAC-seq and ChIP-seq peaks in the Id3 locus revealed accessibility of HE1 and HE2 in Rag2^-/-^, WT DN3, and WT DN4 cells. However, accessibility of HE1 and HE2 was dampened in HEB cKO cells, especially at the DN3 stage, suggesting that HEB may be involved in remodeling the Id3 locus, resulting in a poised state that enables TCR-dependent transcription factors to induce Id3 proportionally to TCR signal strength. These data are now presented as a new “Figure 7 – figure supplement 1” with corresponding Results, Discussion, and Methods updates.

Our next story will be focused on a finer dissection of the Id3 cis-regulatory elements and their combinatorial regulation by HEB/E2A and other transcription factors, and how they relate to specific signaling pathways. For this study, we will modify the language regarding Egr2 to reflect the open questions that still remain to be addressed.

(2) E2A expression is not affected in HEB-deficient cells, raising the question of partial compensation, a point that should be specifically discussed.

This confounding factor is always an issue with E proteins. We have now added a section to the discussion that highlights previous literature and relates it to our findings.

(3) All experiments are done at E18, when fetal gdT17 development predominates. The discussion could address whether these mechanisms extend to neonatal or adult gdT17 subsets.

In our 2017 paper (PMID 29222418) we showed that HEB cKO mice have defects in the production of functional gdT17 cells in fetal and neonatal thymus and in the adult periphery (in lungs and spleen). While the adult thymus does not support the development of fully functional innate gd T cells, it does contain gdTCR+ cells that have activated the Sox-Maf-Rorc network (Yang 2023, PMID 37815917). It will be very interesting to assess the impact of HEB loss on these cells, and we are actively pursuing this goal. For now, we will add a paragraph to the discussion addressing what we know from previous work and what is yet to be learned.

**Reviewer #3 (Public review):**
Summary:The authors of this manuscript have addressed a key concept in T cell development: how early thymus gd T cell subsets are specified and the elements that govern gd T17 versus other gd T cell subsets or ab T cell subsets are specified. They show that the transcriptional regulator HEB/Tcf12 plays a critical role in specifying the gd T17 lineage and, intriguingly, that it upregulates the inhibitor Id3, which is later required for further gd T17 maturation.Strengths:The conclusions drawn by the authors are amply supported by a detailed analysis of various stages of T cell maturation in WT and KO mouse strains at the single cell level, both phenotypically, by flow cytometry for various diagnostic surface markers, and transcriptionally, by single cell sequencing. Their conclusions are balanced and well supported by the data and citations of previous literature.Weaknesses:I actually found this work to be quite comprehensive. I have a few suggestions for additional analyses the authors could explore that are unrelated to the predominant conclusions of the manuscript, but I failed to find major flaws in the current work.I note that HEB is expressed in many hematopoietic lineages from the earliest progenitors and throughout T cell development. It is also noteworthy that abortive gamma and delta TCR rearrangements have been observed in early NK cells and ILCs, suggesting that, particularly in early thymic development, specification of these lineages may have lower fidelity. It might prove interesting to see whether their single-cell sequencing or flow data reveal changes in the frequency of these other T-cell-related lineages. Is it possible that HEB is playing a role not only in the fidelity of gdT17 cell specification, but also perhaps in the separation of T cells from NK cells and ILCs or the frequency of DN1, DN2, and DN3 cells? Perhaps their single-cell sequencing data or flow analyses could examine the frequency of these cells? That minor caveat aside, I find this to be an extremely exciting body of work.

Excellent question, and the underlying answer is yes, loss of HEB renders the cells more open to divergence to non-T lineages, even at the DN3 stage. Although our datasets did not reveal those cells, we have examined this question previously. In our 2011 paper (Braunstein, 2011, PMID 21189289) where we identified “DN1-like” cells arising from HEB-/- DN3 cells in OP9-DL1 co-cultures. These cells responded to IL-15 and IL-7 by differentiating into cytotoxic NK-like cells. We did not detect TCRb rearrangements but did not look for gdTCR rearrangements. Subsequently, multiple papers from other labs showed that ILC2 were greatly expanded in the thymus using Id-overexpression transgenic mice and HEB/E2A-double deficient mice (Miyazaki, 2023, PMID 28514688; Miyazaki, 2025, PMID 39904558; Berrett, 2019, PMID 31852728; Qian, 2019, PMID 30898894; Peng, 2020, PMID:32817168). The ILCs in these mice had TCRg rearrangements, consistent with a shared origin with WT thymic-derived ILCs. In unpublished data from our lab, we found an increase in the numbers of ILC2 but not ILC3 in HEB cKO fetal thymic organ cultures. We did not follow up on this work any further since the topic was being heavily pursued in other labs, but remain very interested in this branchpoint, and will mention the literature in the discussion.

**Joint recommendations for the authors:**
(1) Experimental validation (for mechanistic clarity)The ChIP-seq reanalysis indicates overlapping HEB, E2A, and Egr2 peaks ~60 kb upstream of Id3. Given that the Egr2 data are not generated using the same thymocyte subsets, some form of validation should be considered for the co-binding of HEB and Egr2, potentially ChIP-qPCR in sorted gdT17 progenitors to substantiate the proposed cooperative mechanism.

See above; new experiments with ATAC-seq and additional ChIP-seq analysis.

(2) FiguresPotential inconsistencies in Figure 1H: In the legend to Figure 1H, Vg1-Vg5- cells are considered Vg6+ cells. Flow plots show reduced A Vg1-Vg5- population in HEBc ko mice, but the accompanying bar plot shows increased frequency of Vg6+ cells.

Vg6 cells are actually considered to be Vg4-Vg5-Vg1- cells (not Vg4- Vg1- cells, which is important in the fetal context). The flow plot shows the percentage of Vg6 cells out of the Vg1-Vg4- population, whereas the bar plot shows the percentage of Vg6 cells out of all gdTCR+ cells. The ratio of Vg6 to Vg5 cells decreases within the Vg1-Vg4- population, whereas the overall percentages and numbers of Vg6 cells in all gd T cells is increased in HEB cKO mice. We have now more clearly explained this in the text and the figure legend.

Clarify which cells produce IL-17A in Figure 1L.

This plot is gated on all gd T cells stimulated with PMA/ionomycin; this has been added to the results and figure legend.

In Supplementary Figure 2, legend, do the authors mean that TRGV4 was depleted? The authors write TRDV4. Please check.

Thank you for catching this mistake, we have corrected it.

In Figure 7, the Author showed Id3 mRNA expression. Can the expression of Id2 be included?

That is a really interesting question, and we will follow up on it in future studies.

If Id1 or Id4 are relevant for any of these studies, can their expression be shown in Supplementary Figure 3A? If these are minimally expressed or not expressed, this could be mentioned.

Id1 and Id4 were not detectable in our studies, this is now stated in the results section describing expression of E proteins and Id proteins.

(3) DiscussionDiscuss possible redundancy between HEB and E2A, as E2A expression appears unaffected in HEB-deficient cells.

See above

Address whether the mechanisms identified at E18 (embryonic stage) also apply to neonatal or adult γδT17 subsets.

See above

Expand on how HEB function may relate to other hematopoietic or early lymphoid lineages (NK/ILC, DN1-DN3 stages), based on reviewer curiosity.

See above

(4) Methods and terminologyDefine the terms γδTe1 and γδTe2 (e.g., early effector subsets).

This has been defined more clearly in several sections of the text.

Add details to the scRNA-seq methods section (average number of cells analyzed and sequencing depth per cell).

These details have been added.